# Single-molecule imaging of transcription dynamics, RNA localization and fate in human T cells

M Valeria Lattanzio[1,2,3], Nikolina Šoštarić [ID][1,2,3,5], Nandhini Kanagasabesan[1,2,3], Branka Popović[1,2,3], Antonia Bradarić[1,2,3], Leyma Wardak[1,2,3], Aurélie Guislain[1,2,3], Philipp Savakis[4], Evelina Tutucci [ID][4 ✉] & Monika C Wolkers [ID][1,2,3 ✉]

## Abstract

**T cells are critical effector cells counteracting infections and malignancies. To achieve this, they produce pro-inflammatory cytokines, including IFN-γ and TNF. Cytokine production is a tightly regulated process, but the relative contribution of transcriptional and post-transcriptional regulation to mRNA expression remains unknown. We optimized single-molecule FISH for primary human T cells (T-cell smFISH) to simultaneously quantify nascent RNA, levels of mature mRNA, and its localization with single-cell resolution. T-cell smFISH uncovered heterogeneous cytokine mRNA levels, with high cytokine producers displaying biallelic *IFNG/TNF* RNA transcription activity. Throughout activation, nuclear cytokine mRNAs accumulated, whereas cytoplasmic cytokine mRNA was degraded through translation-dependent decay. Lastly, T-cell smFISH uncovered cytokine-specific regulation by the RNA-binding protein HuR. Thus, T-cell smFISH provides novel insights in the intricate (post)-transcriptional processes in T cells.**

**Keywords** Post-Transcriptional Regulation; smFISH; T Cells; Cytokine Regulation
**Subject Categories** Chromatin, Transcription & Genomics; Immunology; RNA Biology

## Introduction

CD8[+] T cells play a crucial role in clearing infected and malignant cells. Upon target cell recognition, effector CD8[+] T cells undergo a substantial remodeling of their transcriptome and proteome (Araki et al, 2017; Wolf et al, 2020; Rak et al, 2021; Nicolet and Wolkers, 2022). This remodeling allows for the rapid release of effector molecules such as granzymes and pro-inflammatory cytokines (Salerno et al, 2017; Popović et al, 2023). Key pro-inflammatory cytokines are IFN-γ and TNF, which license T cells to kill their target cells (Ikeda et al, 2002; Zhang et al, 2008; Nathan et al, 1983).

Cytokine production is regulated at multiple molecular levels, including transcriptional and epigenetic mechanisms (Conley et al, 2016; Henning et al, 2018). Another important layer of gene expression control is post-transcriptional regulation (PTR), which includes processes such as RNA splicing, RNA transport, RNA stability, and translation control. The importance of PTR is evidenced by the fact that measured mRNA levels in T cells only mildly correlate with protein abundance (Nicolet and Wolkers, 2022), which is a highly conserved feature throughout evolution (Schwanhüusser et al, 2011; Vogel and Marcotte, 2012). For instance, memory T cells contain ready-to-deploy cytokine mRNA, which is blocked through translational control, yet becomes released from this block to serve as a template for translation for rapid recall responses (Salerno et al, 2018). Furthermore, cytokine overproduction of effector T cells induces immunopathology and autoimmunity, and loss of production results in dysfunctional antitumor responses (Karki et al, 2021; Zhang et al, 2008). It is therefore of utmost importance to decipher how PTR processes control T cell function.

RNA-binding proteins (RBPs) are key mediators of PTR that define the abundance of mRNA and protein in T cells (Hoefig et al, 2021; Perez-Perri et al, 2018; Popović et al, 2023). For instance, the RBP ELAVL1/HuR promotes early cytokine production of activated human CD8[+] T cells (Popović et al, 2023). In other cell types, HuR was shown to mediate pre-mRNA splicing, control polyadenylation, regulate the nuclear export of mRNA, and, when located in the cytoplasm, promote mRNA stability and translation (Deka and Saha, 2020; Diaz-Muñoz et al, 2015; Osma-Garcia et al, 2021; Poganik et al, 2019; Tiedje et al, 2012). Provided that RBP expression and their mode of action is cell-type specific (Salerno et al, 2020; Zandhuis et al, 2021), it remains to be uncovered how HuR promotes cytokine production in T cells.

The processing of an RNA molecule occurs at different subcellular locations (Das et al, 2021). In the nucleus, the RNA is transcribed, spliced, and modified. When translocated into the cytoplasm, translation can take place. Previous reports that studied

[1]Sanquin Blood Supply Foundation, Department of Research, T cell differentiation lab, Plesmanlaan 125, Amsterdam, The Netherlands. [2]Landsteiner Laboratory, Amsterdam Institute for Infection & Immunity, Cancer center Amsterdam-Cancer Immunology, Amsterdam UMC, University of Amsterdam, Meibergdreef 9, Amsterdam, The Netherlands. [3]Oncode Institute, Utrecht, The Netherlands. [4]Systems Biology Section, A-LIFE department, Amsterdam Institute of Molecular and Life Sciences (AIMMS), Vrije Universiteit Amsterdam, De Boelelaan 1085, NL-1081HV Amsterdam, The Netherlands. [5]Present address: Department of Bionanoscience, Kavli Institute of Nanoscience, Delft University of Technology, Van der Maasweg 9, 2629 HZ Delft, The Netherlands. ✉E-mail: evelina.tutucci@vu.nl; m.wolkers@sanquin.nl

the regulation of mRNA expression in T cells measured mRNA transcription, mRNA kinetics, and their subcellular localization by bulk RNA-sequencing (Davari et al, 2017; Radford et al, 2016). However, these studies lack subcellular and single-cell resolution, which is crucial to understand the role of posttranscriptional gene regulation during T cell responses, given their inherently high heterogeneity (Nicolet et al, 2017; Han et al, 2012). Single-molecule fluorescence in situ hybridization (smFISH) provides such in-depth insights (Femino et al, 1998; Raj et al, 2008). smFISH has been employed to visualize and quantify—with single-cell resolution—nascent RNAs at the transcription sites (Fang et al, 2013; Senecal et al, 2014), quantify mature mRNA (Bushkin et al, 2014; Franchini et al, 2019; Ma and Mayr, 2018), or to identify its subcellular localization (Chouaib et al, 2020; Maekiniemi et al, 2025). While highly informative, these studies did not simultaneously measure de novo transcription, the level and fate of mRNA in T cells. However, for deciphering gene regulation, integrating this information is paramount. Moreover, studies using smFISH in T cells (Fang et al, 2013; Bushkin et al, 2015) generally employ standard smFISH analysis pipelines that were developed for adherent cells (Eliscovich et al, 2017; Mueller et al, 2013). As these analysis pipelines are not geared towards the small cell size and round morphology of T cells, they fail to unequivocally call the subcellular localization of transcripts. To address these challenges and to comprehensively study cytokine RNA dynamics from transcription to translation, we optimized smFISH for T cells (T-cell smFISH). This optimized analysis pipeline enables the simultaneous quantification of nascent RNA, mature mRNA and the localization of nuclear and cytoplasmic mRNA, overcoming the limitations of the T cell morphology. Using this pipeline, we achieved an unprecedented depth of analyzing the fate of mRNA in T cells, here exemplified for cytokine mRNAs.

# Results

## 3D Quantification of single cytokine mRNA in primary T cells

To quantify cytokine mRNAs in human T cells, we first optimized the smFISH hybridization and analysis protocol (see Materials and Methods). To this end, we generated effector T cells (Teff) that can rapidly respond to recall responses, by activating human blood-derived CD8$^+$ T cells for 72 h with α-CD3/α-CD28, followed by 7 days of rest in culture medium. Teff cells were subsequently restimulated with α-CD3/α-CD28, fixed, and simultaneously probed for *IFNG* and *TNF* mRNA by two-color smFISH (Fig. 1A) (Maekiniemi et al, 2020; Raj et al, 2008). Using a wide-field fluorescence microscope, Z-stacks were acquired every 200 nm to encompass the entire cell thickness, and to precisely identify the x, y, z position of an mRNA molecule (van Otterdijk et al, 2024). T cell outline was deduced from background fluorescence in the smFISH channel and from differential interference contrast images (DIC; Appendix Fig. S1A). DAPI was used as a proxy for nuclear staining (Appendix Fig. S1A). FISH-quant was used to enumerate total mRNAs (Fig. 1B,C). To identify the transcription sites (TsX), high-intensities signal (1.5x the average intensity of a single mRNA) colocalizing with DAPI staining was used (Appendix Fig. S1B, see Materials and Methods). This approach allowed us to

measure cells with one active TsX (mono-allelic) or two active TsX (bi-allelic) (Appendix Fig. S1C). The average intensity of all mRNAs was used as a reference to quantify the number of nascent mRNA per TsX. To localize individual cytokine mRNA molecules with subcellular resolution, we developed the T-cell smFISH pipeline, which is optimized to account for the small and compact structure of T cells by combining FISH-quant (Mueller et al, 2013) for RNA quantification, filtering for cell selection, and Cell-Pose (Stringer et al, 2020) for 3D mask reconstruction (Appendix Fig. S1D; Movies EV1 and EV2; https://github.com/nikolinasostaric/T-cell_smFISH). T-cell smFISH achieved a 98% accuracy for localization, as defined by manual analysis of 100 randomly chosen mRNAs. T-cell smFISH thus allowed us to elucidate the expression of cytokine transcripts upon T cell activation with single-cell and subcellular resolution.

## T-cell smFISH uncovers cytokine-specific transcription activity

Having established T-cell smFISH, we studied the cytokine (m)RNA expression changes during T cell activation with α-CD3/α-CD28. The key effector cytokines IFN-γ and TNF are produced early after T cell activation, yet with distinct kinetics (Nicolet et al, 2017; Han et al, 2012). To uncover the underlying mechanism, we measured nascent RNA and the subcellular distribution of mature mRNA (Appendix Fig. S1E), and we compared that to cytokine protein levels.

We first focused on nascent cytokine RNAs. Resting Teff cells (t = 0 h) lacked active transcription sites (TsX) for *IFNG* (Fig. 1D; Appendix Fig. S1F). At 1 h of activation, however, 2% Teff cells expressed nascent *IFNG* RNA with a median of 12 nascent RNAs per TsX (Fig. 1D; Appendix Fig. S1F,G). At 3 h of activation, active TsX peaked with 8%, with a median of three nascent *IFNG* RNAs per TsX, and this level of nascent RNA was maintained up to 6 h of activation (Fig. 1D; Appendix Fig. S1F,G). Of note, even though at 1 h of activation we observed bi-allelic transcription in 45% of the *IFNG*-transcribing cells, at later time points mono-allelic *IFNG* transcription was more prominent (Appendix Fig. S1F).

In contrast to *IFNG*, nascent *TNF* RNA was already detected in 1% of resting Teff cells with a median of five nascent RNA molecules per TsX (Fig. 1D; Appendix Fig. S1F,G). At 1 h of stimulation, nascent *TNF* peaked with 13 nascent RNA per TsX, which dropped and stabilized at 2 h with six molecules on average per cell (Fig. 1D; Appendix Fig. S1F,G). Intriguingly, this drop did not result from fewer active TsX for *TNF*. Furthermore, bi-allelic transcription was observed in 56% of the TNF-transcribing cells until 3 h of activation, and only at later time points, mono-allelic transcription was more prevalent (Appendix Fig. S1F). Thus, nascent *IFNG* and *TNF* RNA display different kinetics, with *IFNG* requiring T cell activation for de novo synthesis, and *TNF* transcription already being active in resting Teff cells. *IFNG* transcription relies more on mono-allelic transcription, whereas *TNF* transcription also employs bi-allelic synthesis during early T cell activation.

## Cytokine-specific kinetics of mRNA and protein expression during T cell activation

We next quantified mature *TNF* and *IFNG* mRNA. Even though nascent *IFNG* RNA was undetectable in resting Teff cells, 36%

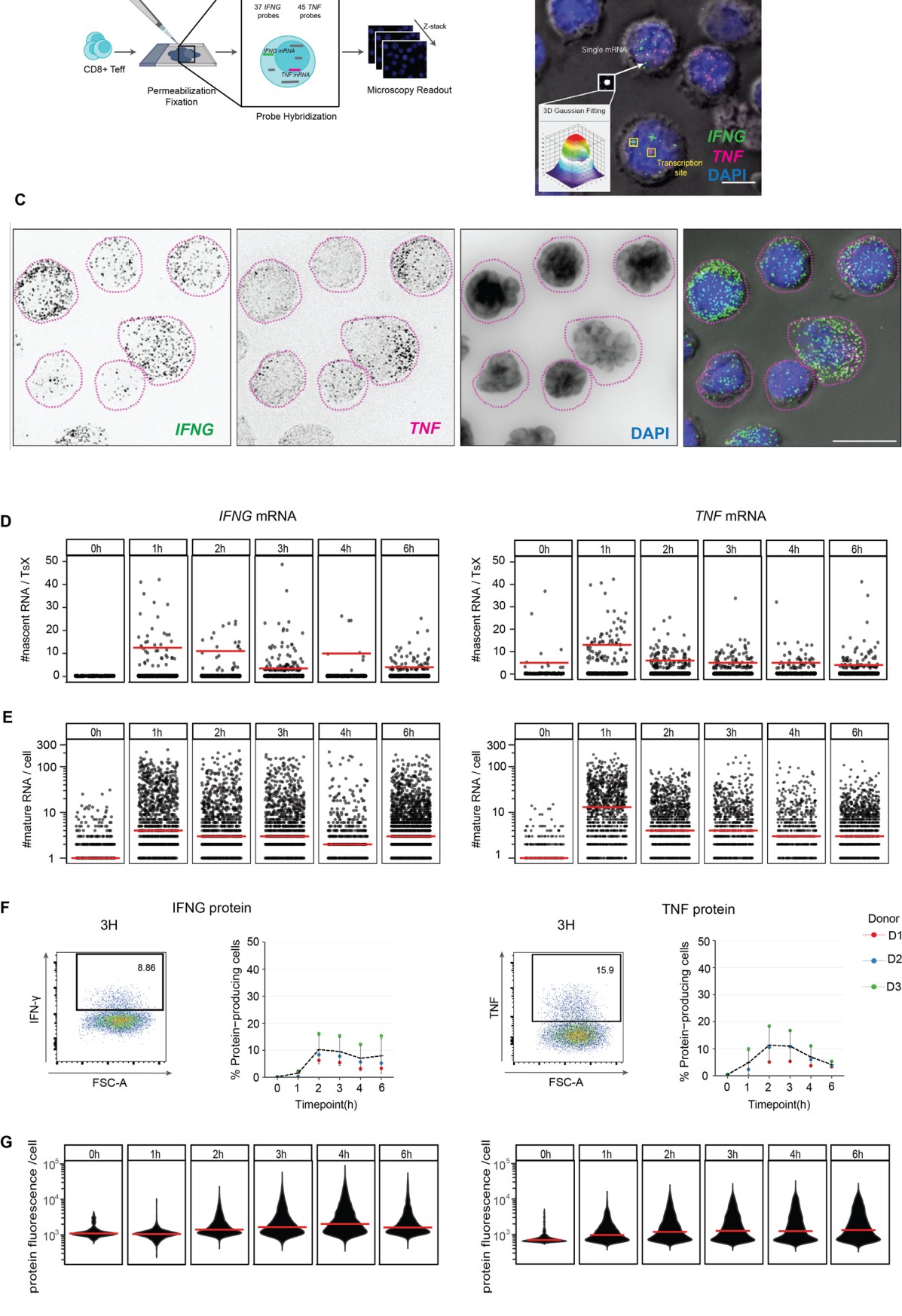

**Figure 1.   T-cell smFISH quantification of *IFNG* and *TNF* mRNA in CD8$^+$ Teff cells.**

(A) Schematic workflow of T-cell smFISH. (B) Maximal projection of smFISH for *IFNG* mRNA (green), *TNF* mRNA (magenta), and DAPI (blue) merged to a single DIC section (gray). Yellow squares indicate *IFNG* and *TNF* transcription sites (TsX). Lower left inset: Gaussian signal of a single mature mRNA. Scale bar: 5 µm. (C) Maximal projection of smFISH for *IFNG* mRNA (first image, green), *TNF* mRNA (second image, magenta), DAPI (third image, blue) and all previous images merged on DIC. Dotted lines represent 2D cell outline. Scale bar: 10 µm. (D) Number of nascent *IFNG* (left) and *TNF* (right) RNA per TsX. Each dot depicts one transcription site. Data pooled from three donors. *n* ≥ 1000 cells/time point. Red bar: median. (E) Number of mature *IFNG* (left) and *TNF* (right) mRNA in Teff cells expressing ≥1 mature mRNA. Each dot represents an individual cell. Data pooled from three donors. (F) Cytokine expression measured by intracellular cytokine staining. BrefA was added for max. 2 h. Left panels: representative flow cytometry plot. Right panels: *n* = 3 donors, line = mean. (G) Violin plot of protein fluorescence intensity per cell. Data pooled from three donors. Red bar: median. Source data are available online for this figure.

expressed low but detectable levels of mature *IFNG* mRNA, and 28% expressed mature *TNF* mRNA (each a median of one molecule/cell; Fig.1E; Appendix Fig. S1H). At 1 h of stimulation, 77 and 74% of the T cells expressed mature *IFNG* and *TNF* mRNA, respectively (median of 4 and 13 molecules/cell, respectively; Appendix Fig. S1H,I). Yet, whereas a subset of Teff cells maintained high levels (>100 molecules/cell) of mature *IFNG* mRNA at all time points measured, the levels of mature *TNF* mRNA rapidly declined from 2 h of activation onwards to below 100 molecules/cell (Fig. 1E). *IFNG* mRNA expression rather displayed a wave-like expression, that decreased at 4 h from the initial induction but then increased again at 6 h poststimulation, as was previously described for its protein expression (Han et al, 2012) (Fig. 1E).

Notably, a high intercellular variability was observed with 1–200 *IFNG* and *TNF* mRNA molecules/cell in all three donors, and at all measured time points (Fig. 1E; Appendix Fig. S1I). This intercellular and inter-donor heterogeneity was also measured for the protein production of IFN-γ and TNF (Fig. 1F), as previously reported (Nicolet et al, 2017; Han et al, 2012). Yet, the overall response kinetics were comparable between donors. Following the expression patterns of mature mRNA, IFN-γ protein production peaked at 2 h of activation with a maximum of 10%, and it remained stable thereafter (Fig. 1F). Furthermore, also at protein levels a high intercellular variability was detected, which on population level peaked at 4 h post stimulation (Fig. 1G, *p* < 0.0001 compared to all other time points). TNF production was more rapid: in line with higher *TNF* mRNA levels at 1 h of activation (Fig. 1E, *p* < 0.0001 compared to 0 h), TNF protein production peaked at 2–3 h of activation with 12% TNF-producing Teff cells, and the percentage rapidly declined thereafter (Fig. 1F). Nevertheless, the protein production per cell remained constant throughout (Fig. 1G). Combined, these data indicate that the protein production kinetics of IFN-γ and TNF follow the cytokine mRNA kinetics.

## Dual cytokine mRNA-expressing Teff cells dominate the immune response

Dual cytokine producers are more potent effector T cells than single cytokine producers in response to infection and cancer (Almeida et al, 2007a; De Groot et al, 2019). We therefore questioned how single (SP) and double positive (DP) cytokine mRNA expressors (Movie EV1) correlate with the observed high variability of cytokine mRNA expression. When enumerating the number of nascent cytokine RNA, significantly higher numbers of nascent *IFNG* RNAs per TsX were measured in DP compared to SP Teff cells (Fig. 2A). This was not the case for nascent *TNF* (Fig. 2A).

In contrast, the percentage of active TsX was substantially higher for both cytokines in DP Teff cells, and this was observed at all time points (Appendix Fig. S2A). In addition, DP T cells displayed more bi-allelic transcription for both cytokines (Appendix Fig. S2A). The higher transcriptional activity in DP T cells also resulted in significantly higher numbers of mature cytokine mRNA per cell (Fig. 2B,C). Upon T cell activation, DP T cells expressed substantially higher percentages and numbers of mature *IFNG* mRNA/cell at all time points measured and mature *TNF* mRNA at early T cell activation time points (1–3 h) compared to SP T cells (Fig. 2B,C). Not only mRNA, but also the percentage of protein-producing Teff and protein production/cell was higher in DP cytokine mRNA producers (Fig. 2D; Appendix Fig. S2B). In conclusion, Teff cells with active TsX for both *IFNG* and *TNF* contain more mature cytokine mRNA, and the higher mRNA levels originate from higher levels of bi-allelic transcription. Furthermore, DP cytokine producers dominate the cytokine production kinetics upon T cell activation and contribute to the observed high intercellular heterogeneity of cytokine production.

## Differential subcellular distribution of *IFNG* and *TNF* mRNA

Mature cytokine mRNA can only serve as a template for translation when located in the cytoplasm. Therefore, we next investigated the subcellular distribution of *IFNG* and *TNF* mRNA during activation with T-cell smFISH (Appendix Fig. S3A,B). In resting Teff cells, the vast majority of *IFNG* mRNA and *TNF* mRNAs were present in the nucleus (Fig. 3A; Appendix Fig. S3A—left panel, S3B —top panel). However, already 1 h of T cell activation rapidly shifted the distribution of mature *IFNG* mRNA towards the cytoplasm (Fig. 3A), indicating rapid translocation after T cell activation. From 2 h of activation onwards, the *IFNG* mRNA was equally distributed between nucleus and cytoplasm (Fig. 3A). *TNF* mRNA was more prevalent in the nucleus throughout the activation phase, which was with 75% nuclear mRNA most apparent at the peak of its expression, i.e., at 1 h of T cell activation (Fig. 3A; Appendix Fig. S3A,B). Thus, mature *IFNG* and *TNF* mRNAs differentially distribute between the nucleus and cytoplasm in activated Teff cells.

## Translation-dependent decay controls cytokine mRNA levels upon T cell activation

The unexpectedly high levels of mature cytokine mRNA in the nucleus prompted us to question how de novo transcription and translation contributed to the subcellular distribution. We therefore activated Teff cells for 2 h, and in the second hour of activation, we

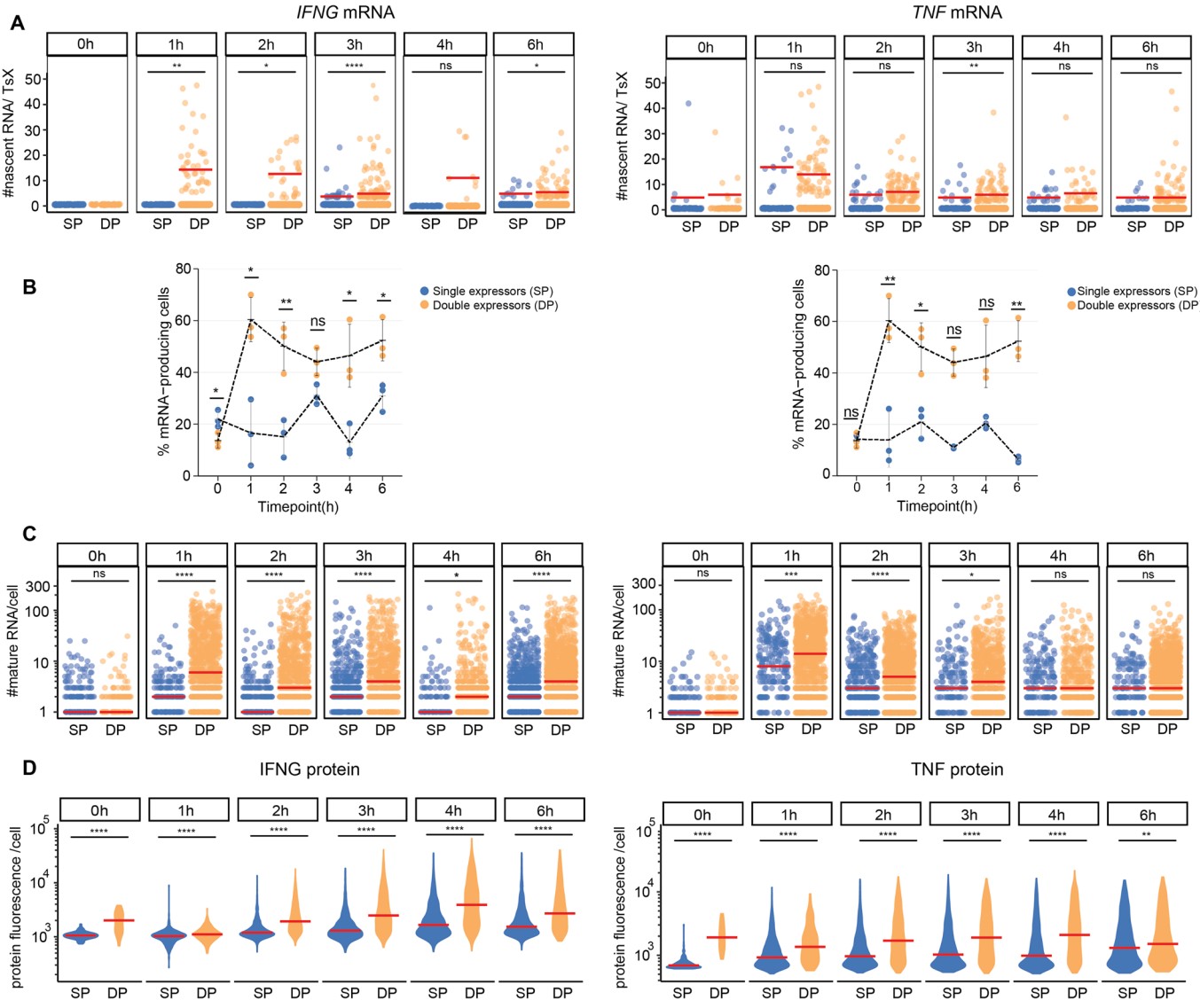

**Figure 2. Cytokine mRNA and protein expression is dominated by double expressors.**

(A) Number of nascent *IFNG* (left) and *TNF* (right) RNA per TsX in single positive (SP, blue) or double positive (DP, orange) expressors upon T cell activation. Each dot represents an individual cell, data pooled from three donors. Red bar: median. (B) Percentage of Teff cells expressing ≥1 mature *IFNG* (left) and *TNF* (right) mRNA in single and double positive mRNA expressors. $n = 3$ donors, line: mean. (C) Number of mature *IFNG* (left panel) and *TNF* (right panel) mRNA SP and DP expressors containing ≥1 mRNA. Each dot represents an individual cell. Data pooled from three donors. Red bar: median. (D) Violin plot of protein fluorescence intensity of cytokine production in SP and DP producers per cell. Data pooled from three donors. Red bar: median. *$p ≤ 0.05$, **$p ≤ 0.01$, ***$p ≤ 0.001$, ****$p ≤ 0.0001$ ns: non-significant. Kruskal–Wallis non-parametric test, and post hoc Tukey-HSD test for comparing time points. For exact $p$ values, see Dataset EV1. Source data are available online for this figure.

blocked either de novo transcription with Actinomycin D (ActD), or we blocked translation with Harringtonine (Harr) (Fig. 3B). As expected, Harr treatment reduced both the overall translation as defined by puromycin incorporation, and the cytokine production (Appendix Fig. S3C,D). Likewise, T-cell smFISH analysis confirmed that ActD almost completely blocked the presence of nascent RNA (Fig. 3C). ActD treatment also substantially reduced the number of mature *IFNG* mRNA molecules/cell (Fig. 3D). Intriguingly, the mRNA expression of mature *TNF* was much less affected by ActD treatment (Fig. 3D). Because *TNF* mRNA was not found to be stabilized upon T cell activation (Popović et al, 2023), the data suggest that the production boost of *TNF* mRNA we observed

during the first hour of T cell activation (Fig. 1E) suffices to maintain high levels of mature *TNF* mRNA expression during the second hour of activation.

Notably, blocking translation with Harr increased the number of mature cytokine mRNA/cell compared to control (Fig. 3D, $p < 0.0001$). T-cell smFISH uncovered that Harr treatment primarily increased the cytoplasmic *IFNG* mRNA and *TNF* mRNA (Fig. 3E, $p < 0.0001$), suggesting that the effect of Harringtonine on cytokine mRNA expression resulted from blocking translation. Previous studies reported that the cytoplasmic RNA turnover can be influenced by translation (Horvathova et al, 2017; Tuck et al, 2020). To determine whether the observed cytoplasmic

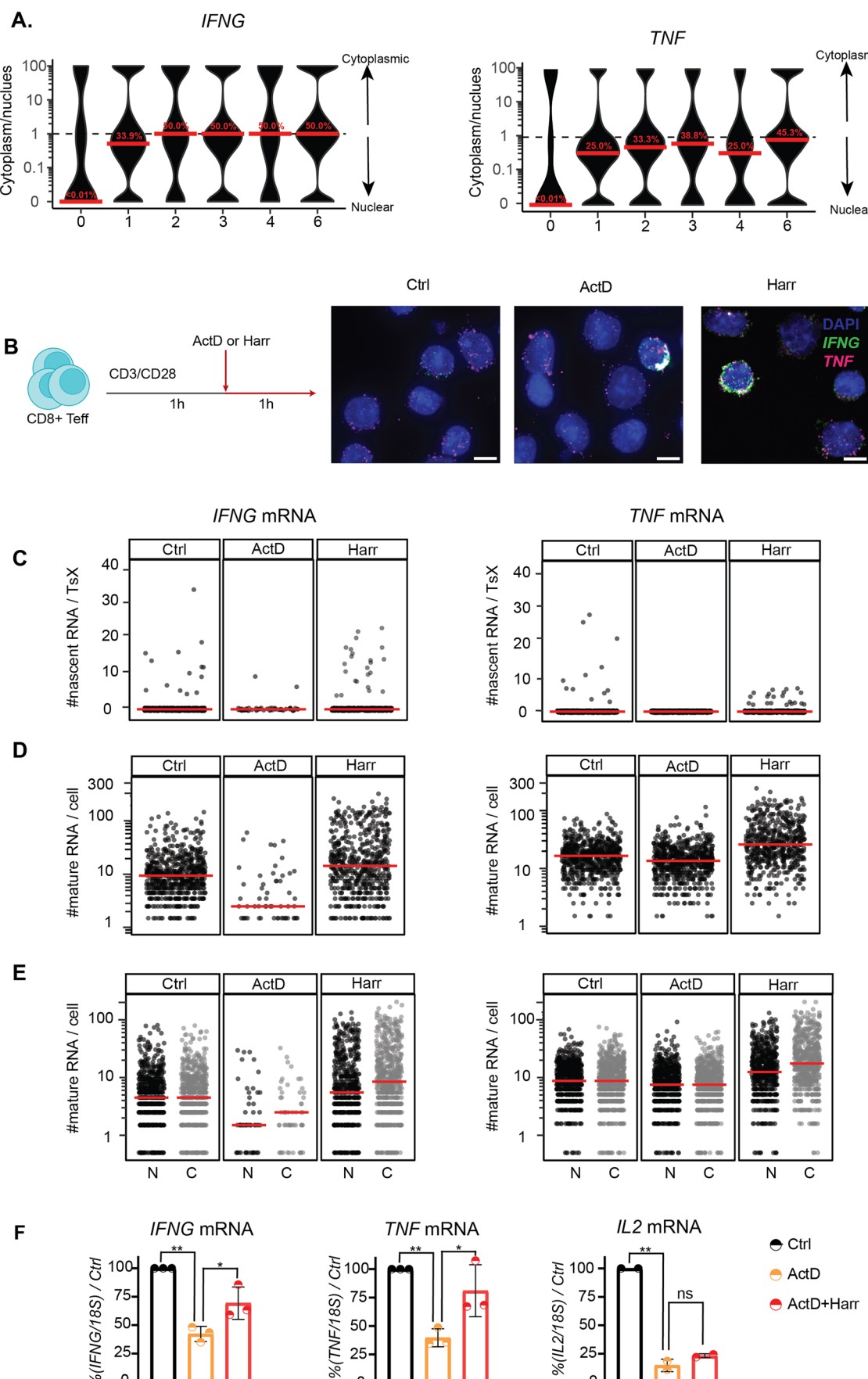

**Figure 3.  *IFNG* and *TNF* mRNA localization and translation control.**

(A) Ratio of cytoplasmic mRNA versus nuclear mRNA per cell of Teff cells expressing ≥1 mature mRNA. Data represented as pseudo-log10 transformed. The dotted line represents equal distribution in the nucleus and cytoplasm. n ≥ 500 cells/time point, pooled from three donors. Red bar: median. Percentage in red: median of the percentage of mRNA in the cytoplasm. (B) Left: Experimental setup of transcription/translation block in Teff cells. Right: MAX projections of T-cell smFISH. Scale bar: 5 μm. (C) Number of nascent *IFNG* (left) and *TNF* (right) RNA per TsX. Each dot represents an individual cell. n ≥ 1000 cells/condition, one donor. Red bar: median. (D) Mature cytokine mRNA of Teff cells expressing ≥1 mature mRNA. (E) Number of nuclear (N, black) and cytoplasmic (C, gray) cytokine mRNA. (F) Cytokine mRNA expression measured by qRT-PCR upon treatment with Actinomycin D (ActD), or ActD for 20 min, then combined with Harringtonine (Harr), normalized to control (Ctrl). n = 3 donors, mean ± SD, p ≤ 0.05, **p ≤ 0.01, ***p ≤ 0.001, ****p ≤ 0.0001 ns non-significant. Two-tailed ratio unpaired Student's t-test. For exact p values, see Dataset EV1. Source data are available online for this figure.

accumulation of cytokine mRNA in T cells upon Harr treatment resulted from translation-dependent mRNA decay, we first blocked transcription and subsequently translation and measured cytokine mRNA by qRT-PCR. ActD treatment alone reduced the overall cytokine mRNA levels by 60% compared to control (Fig. 3F, $p < 0.01$), however, this reduction had little effect on the protein output (Appendix Fig. S3E,F). In contrast, ActD/Harr treatment combined substantially reduced the protein production (Appendix Fig. S3E,F). Yet, the expression levels for *IFNG* and *TNF* mRNA increased by 30% in ActD/Harr-treated Teff cells compared to Act treatment alone ($p < 0.05$), suggesting that the availability of *IFNG* and *TNF* mRNA for translation is regulated by translation. Of note, this increase was not observed for *IL2* mRNA, which suggests transcript specificity for this process (Fig. 3F). Combined, the T-cell smFISH pipeline uncovered—with single-cell resolution—that translation regulates the cytoplasmic *IFNG* and *TNF* mRNA levels. Furthermore, *IFNG* and *TNF* mRNA—but not *IL2* mRNA—are subject to translation-dependent decay.

## HuR differentially controls *IFNG* and *TNF* mRNA expression and cytokine production

Having established the cytokine mRNA expression kinetics and distribution in Teff cells, we questioned how RNA-binding proteins such as HuR regulate the fate of cytokine mRNA. We previously showed that HuR promotes cytokine production during the early phase of T cell activation, i.e., at 1–2 h, but not at later time points (Popović et al, 2023), Thus, HuR activity supports the rapid response rate of effector T cells. Its mode of action is, however, unresolved. HuR protein expression increased in Teff cells during the 2 h of activation (Appendix Fig. S4A). However, RNA-immunoprecipitation (RIP)-qRT-PCR revealed that HuR only interacted with *IFNG* and *TNF* mRNA at 1 h poststimulation, and not at 2 h poststimulation (Appendix Fig. S4B). To study its mode of action on cytokine mRNA, we depleted HuR from Teff cells by CRISPR-Cas9 gene editing (Appendix Fig. S4C). As previously reported (Popović et al, 2023), HuR-KO Teff produced less IFN-γ and TNF protein during the first 2 h of activation, both in terms of percentage and of protein production/cell (Fig. 4A,B). This is also observed when individual donors are plotted (Appendix Fig. S4D). Notably, T-cell smFISH analysis uncovered opposite effects on mRNA levels (Fig. 4C, D). Resting HuR-KO Teff cells already contained a higher percentage of *IFNG* mRNA-expressing cells than control-treated T cells (Appendix Fig. S4E). The median mRNA molecules/cell increased from 3 to 10 for *IFNG*, and from 5 to 9 for *TNF* mRNA (Fig. 4D). At 1 h of stimulation, the percentage of mRNA-expressing Teff cells and the number of cytokine molecules/cell was also higher in HuR-KO (Fig. 4D; Appendix

Fig. S4E). Only at 2 h activation, the number of cytokine mRNAs/cell dropped below that of control Teff cells (Fig. 4D).

To determine whether the higher cytokine mRNA levels observed in HuR-KO Teff cells resulted from increased transcription, we measured the number of nascent RNA. Resting HuR-KO Teff cells had, on average, eight nascent *IFNG* RNA/cell compared to five nascent *IFNG* RNA/cell in control T cells, and this slight increase was lost by 2 h postactivation (Fig. 4E). In contrast, nascent *TNF* RNA remained unaffected by HuR depletion at all time points (Fig. 4E). Thus, despite that HuR-KO T cells produce less cytokine during early activation, mature mRNAs accumulate, and transcription may only partly contribute to this accumulation.

## HuR deletion affects the cellular distribution of *IFNG* and *TNF* mRNA

Because higher cytokine mRNA levels in HuR-KO cells did not result in higher but lower protein production, we questioned whether HuR depletion affected the subcellular mRNA distribution. T-cell smFISH analysis uncovered that resting HuR-KO Teff cells expressed significantly more cytoplasmic *IFNG* and *TNF* mRNA molecules than control Teff cells (Fig. 5A; Appendix Fig. S5A). HuR-KO Teff cells also contained slightly more nuclear *TNF* mRNA (Fig. 5A; Appendix Fig. S5A). Upon 1 h activation, both cytokine mRNAs were slightly increased in both subcellular compartments, which was reversed at 2 h of T cell activation (Fig. 5A; Appendix Fig. S5A). When we determined the relative distribution of cytokine mRNA within each individual Teff cell, we found that resting HuR-KO T cells contain more *IFNG* mRNA in the cytoplasm, which drops at 1 h but is reverted at 2 h poststimulation (Fig. 5B). In contrast, *TNF* mRNA is slightly more nuclear at 0 h and 1 h of activation in HuR-KO cells compared to control cells, and this effect is lost at 2 h of activation, at a time point when HuR also loses its interaction with the cytokine mRNAs (Fig. 5B; Appendix Fig. S4B). In conclusion, HuR deletion influences the subcellular localization of cytokine mRNAs, and it does so in a time-dependent manner.

## HuR regulates the polyadenylation of *TNF* mRNA

Lastly, we aimed to decipher how HuR regulates cytokine RNA. HuR can stabilize its target mRNAs (Rothamel et al, 2021; Wang et al, 2000). However, overall mRNA levels at 1 h of activation were not altered (Fig. 5C). Furthermore, measuring mRNA decay in T cells activated for 1 h, and then treated for an additional 1–2 h with ActD showed no differences in RNA decay rates between HuR-KO and control Teff cells (Fig. 5C). Thus, HuR does not appear to regulate cytokine mRNA stability.

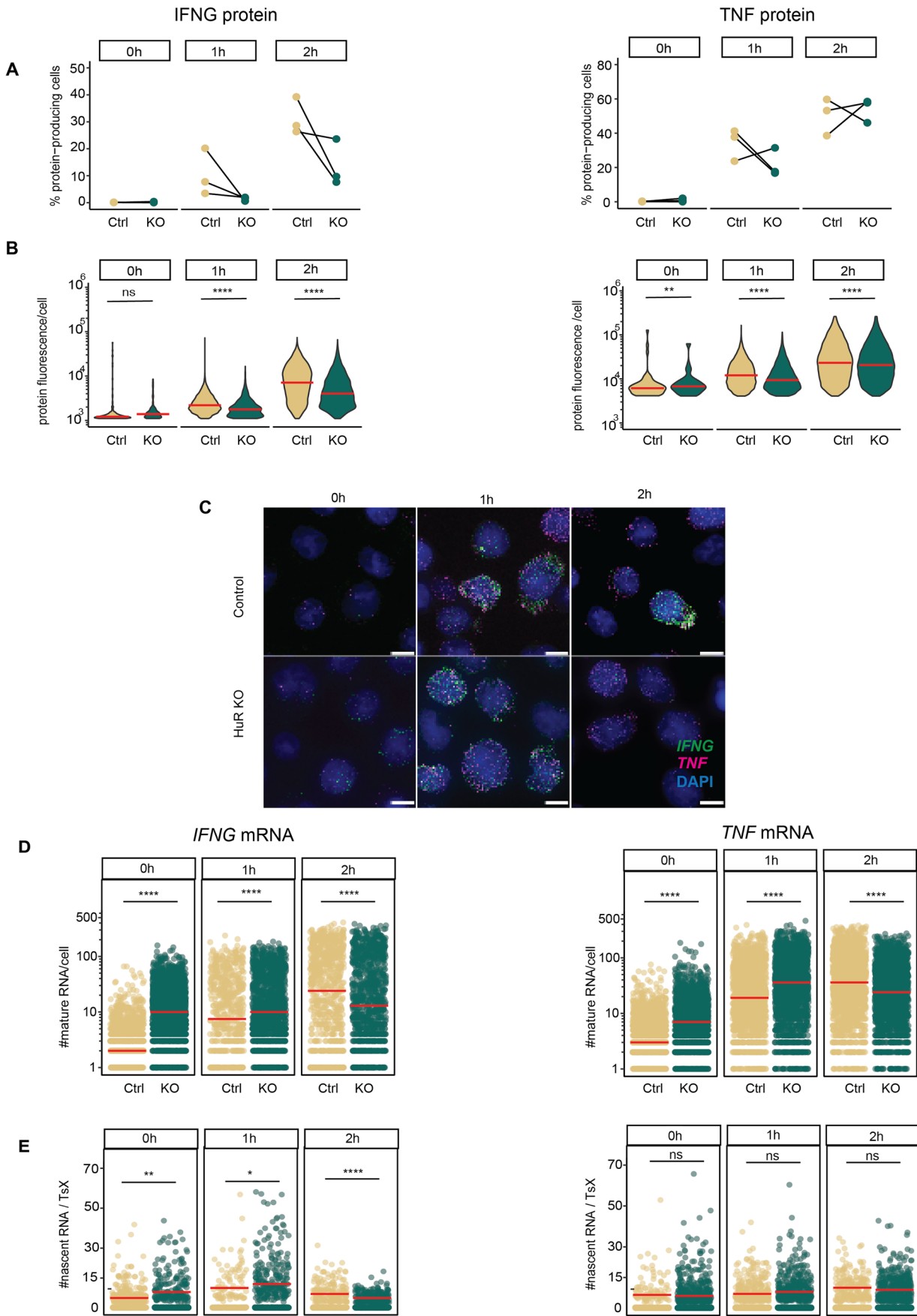

**Figure 4.** **IFNG and TNF mRNA and protein expression in HuR-KO Teff cells.**

(A) Percentage of cytokine-producing Ctrl (C) and HuR-KO Teff cells (KO). BrefA was added at 0 h. $n = 3$ donors. ns non-significant. Kruskal–Wallis non-parametric test, and post hoc Tukey-HSD test. (B) Violin plot of protein fluorescence intensity per cell, from three pooled donors. Red bar: median. (C) Control (top panel) and HuR-KO (bottom panel) Teff cells were activated with α-CD3/α-CD28. MAX projection of T-cell smFISH. Scale bar: 5 μm. (D) Number of mature cytokine mRNA in Teff cells expressing ≥1 mature mRNA. $n \geq 1500$ cells/time point, from three pooled donors. Red bar: median. (E) Number of nascent cytokine RNA per TsX. Red bar: median. *$p \leq 0.05$, **$p \leq 0.01$, ***$p \leq 0.001$, ****$p \leq 0.0001$ ns non-significant. Kruskal–Wallis non-parametric test, and post hoc Tukey-HSD test for comparing time points. For exact $p$ values, see Dataset EV1. Source data are available online for this figure.

HuR can also act as a splicing factor (Akaike et al, 2014; Diaz-Muñoz et al, 2015). Yet, irrespective of the nuclear *IFNG* mRNA accumulation (Fig. 5B), no overt differences of intron-exon junctions were found between HuR-KO and control Teff cells (Appendix Fig. S5B). Additionally, even though splicing is considered a key event for *TNF* (Namer et al, 2017; Yang Yang et al, 1998), HuR deletion has only limited—if any—effect on *TNF* splicing (Appendix Fig. S5B), indicating that HuR is not essential for *IFNG* and *TNF* RNA splicing in T cells. Lastly, HuR can regulate polyadenylation of its target mRNA (Deka and Saha, 2020; Poganik et al, 2019). For *IFNG* mRNA, we found no changes in the poly(A) tail profile in HuR-KO cells at 1 h of T cell activation (Fig. 5D; Appendix Fig. S5C). In sharp contrast, HuR deficiency results in changes of the polyadenylation profile of *TNF* mRNA, as defined by the RNA ligation-mediated poly(A) test (RL-PAT), which measures the poly(A) tail length of mRNAs a (Meijer et al, 2007). Thus, HuR deficiency impairs the canonical poly(A) tail. (Fig. 5D; Appendix Fig. S5C). In conclusion, HuR employs cytokine-specific regulatory mechanisms to control the expression of mRNA, and ultimately the protein expression of these two key pro-inflammatory cytokines.

## Discussion

In this study, we present the T-cell smFISH pipeline to simultaneously quantify nascent and mature cytokine mRNAs and their subcellular distribution. Importantly, the three-dimensional analysis of RNA expression with single-molecule and single-cell resolution provided new insights into the regulation of cytokine production and the activity of RNA-binding proteins in T cells, as we showcased for HuR.

Our study underscores the distinct temporal kinetics of IFNG and TNF expression during T cell activation. Consistent with previous reports (Nicolet et al, 2017; Han et al, 2012), the TNF production peaks early, followed by IFNG production, which is most prominently produced at 3–6 h post activation —a time point when TNF levels have already declined. These temporal dynamics reflect the distinct functional roles of the cytokines in T cell responses: TNF contributes to early cytotoxic activity and facilitates immune cell recruitment, whereas IFNG plays a central role in sustaining antiviral and antitumor immunity at later stages of T cell activation (Soudja et al, 2014; Griffin et al, 2012).

This tight regulation of cytokine production and its restriction to specific time points is paramount for their physiological function, which is indeed manyfold for both IFN-γ and TNF (Young and Hardy, 1995; Tracey and Cerami, 1994). Therefore, tight regulation of cytokine production is required to prevent avert side effects. We here present how the regulation of mRNA expression contributes to the cytokine expression profile, and uncover cytokine-specific regulatory nodules. Of note, the length

and duration of cytokine productions is also defined by the strength and type of signals a T cell receives (Zehn et al, 2009; Salerno et al, 2016). Furthermore, the response rate is linked to their cellular differentiation status and, as a result, the epigenetic landscape at the TNF and IFNg locus (Denton et al, 2011). It will therefore be interesting to uncover how different signals received by T cells, both antigen-dependent and -independent, shape the fate of cytokine mRNA by T cells and ultimately the cytokine production.

Even though the majority of activated Teff cells expressed cytokine mRNA, the single-cell resolution of smFISH revealed a wide range of mRNA expression (i.e., from one mRNA molecule to hundreds of mRNA molecules per cell; Fig. 1E). This heterogeneity within the T cell population was observed at all time points and was also observed for the protein expression (Fig. 1E,G). Within these heterogeneous T cell responses, cytokine production upon T cell activation primarily originates from Teff cells that transcribe both cytokine mRNAs simultaneously, suggesting that this superior cytokine production by DP cells is cell-intrinsic. The functional heterogeneity in cytokine production we report here aligns well with prior studies emphasizing the relevance of polyfunctional T cells producing more than one cytokine in response to infected and malignant cells (Wimmers et al, 2015; Almeida et al, 2007b; Betts et al, 2006; La Gruta et al, 2004). The molecular mechanisms driving this dual cytokine production remain to be fully elucidated. However, several non-mutually exclusive explanations may contribute to this phenomenon: (1) an open chromatin structure at the cytokine loci or their respective regulators, (2) high expression of cytokine-driving transcription factors, or (3) more responsive signaling cascades in response to T cell activation. Future T-cell smFISH analyses may support the quest to uncover the underlying mechanisms.

T-cell smFISH uncovered that *IFNG* and *TNF* mRNA accumulate in the nucleus, suggesting that nuclear retention regulates the availability of cytokine mRNAs for translation, as previously shown in other cell types (Battich et al, 2015). In addition, nuclear retention of *TNF* mRNA may support its efficient splicing as previously suggested (Namer et al, 2017; Yang Yang et al, 1998). We therefore hypothesize that the subcellular localization of cytokine mRNA dictates the amount of cytokine production. Furthermore, we here uncovered that translation-dependent decay (Mercier et al, 2024) contributes to the lower mRNA levels we measured in the cytoplasm, a feature that appeared transcript-specific, as *IL2* mRNA was resistant.

T-cell smFISH also provides novel insights into the mode of action of RBPs. HuR deficiency resulted in increased levels of *IFNG* and *TNF* mRNA, yet with lower protein production. Of note, smFISH probes can detect mature and pre-spliced RNA (Femino et al, 1998; Mueller et al, 2013). However, splicing of *IFNG* and *TNF* was not affected in HuR-KO T cells. HuR can also promote alternative polyadenylation and compete with the PolyA cleavage factor (Dai et al, 2012; Deka and Saha, 2020). We found that HuR is required for the polyadenylation of *TNF*, but not of *IFNG* mRNA. The functional consequence of altered

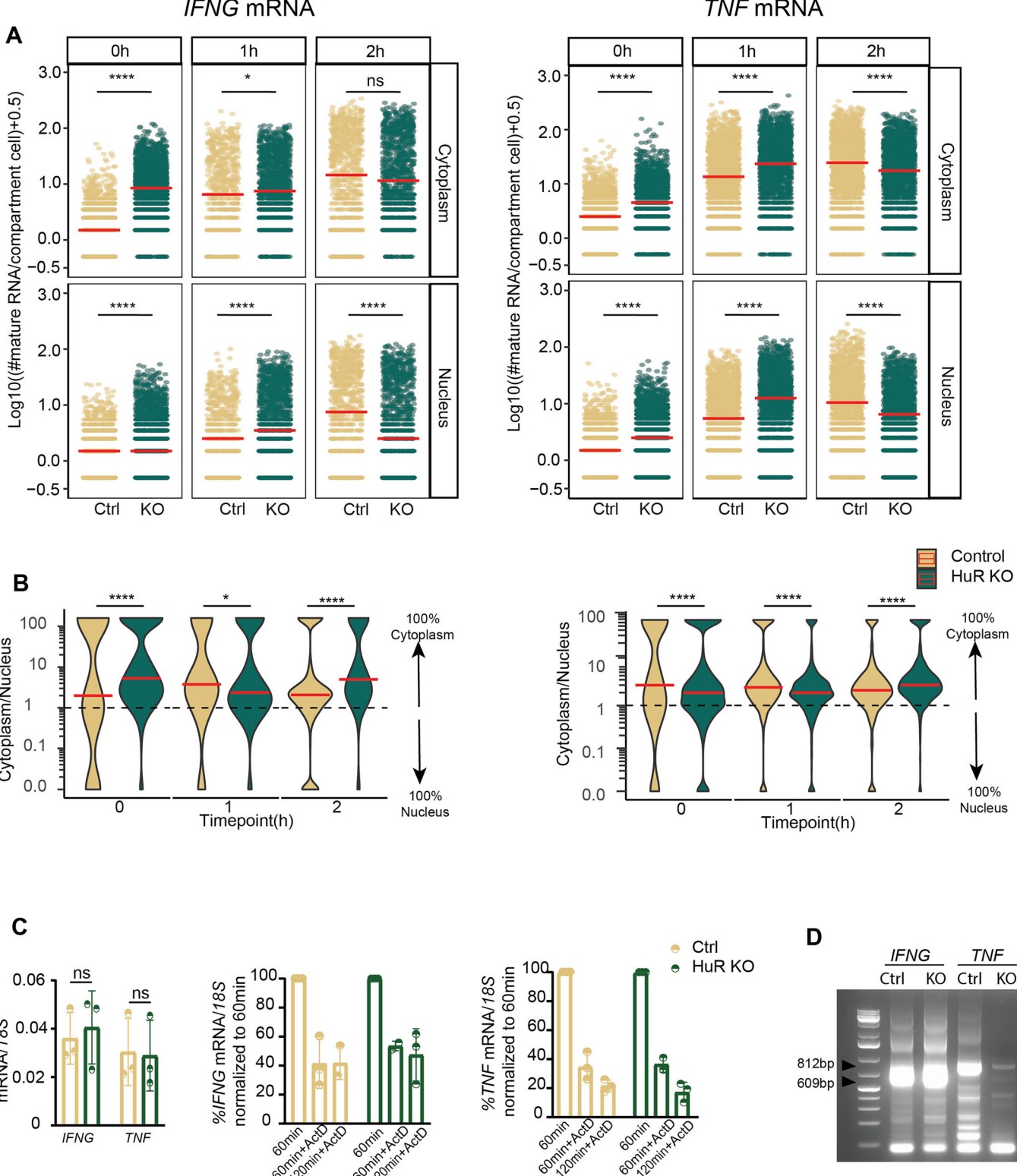

**Figure 5.  *IFNG* and *TNF* mRNA localization and PTR in HuR-del Teff cells.**

(A) Number of cytoplasmic (top) and nuclear (bottom) cytokine mRNAs in control (Ctrl) or HuR-KO (KO) Teff cells that express ≥1 mature mRNA per cell. Data represented as pseudo-log10 transformed, with a coefficient of 0.5 added. Each dot represents an individual cell, data pooled from three donors. Red bar: median. (B) Ratio of cytoplasmic mRNA versus nuclear mRNA in Ctrl or HuR-KO Teff cells expressing ≥1 mature mRNA per cell. Data represented as pseudo-log10 transformed. Data from three pooled donors. Red bar: median. For (A, B) $^*p ≤ 0.05$, $^{**}p ≤ 0.01$, $^{***}p ≤ 0.001$, $^{****}p ≤ 0.0001$ ns non-significant. Kruskal–Wallis non-parametric test, and post hoc Tukey-HSD test for comparing time points. For exact $p$ values, see Dataset EV1. (C) Left: *IFNG* and *TNF* mRNA expression determined by qRT-PCR in Ctrl or HuR-KO Teff cells activated for 1 h with α-CD3/α-CD28. Right *IFNG* and *TNF* mRNA levels in Teff upon ActD treatment. Cytokine mRNA expression at 1 h of T cell activation is considered 100%. $n = 3$ donors, mean ± SD, Data non-significant. Two-tailed ratio paired $t$-test. (D) Measurement of poly(A) length of cytokine mRNA using RL-PAT in control (Ctrl) or HuR-KO (KO) Teff cells activated for 1 h with α-CD3/α-CD28 using the RL-PAT assay. Arrows indicate *IFNG* 3'UTR full-length: 812 bp and *TNF* 3'UTR full-length: 609 bp. Source data are available online for this figure.

polyadenylation remains to be uncovered. Polyadenylation can modulate mRNA stability (Passmore and Coller, 2021), but that was not the case for *TNF*. This finding indicates that the poly(A) tail length for *TNF* is decoupled from mRNA stability and translation output. Of note, such decoupling is not unique to HuR depletion. Previous studies reported that mRNAs, including *TNF*, remained translationally active in activated macrophages, even though the polyA tails were shortened (Kwak et al, 2022; Crawford et al, 1997). Whether this is a feature unique to immune cells, or whether it generally occurs in cells that rapidly adapt to external cues remains to be determined. How polyA tail length is uncoupled from mRNA abundance and translation is to date unresolved. It is conceivable that secondary structures in the mRNA or RBPs interacting with *TNF* mRNA impede access to the degradation machinery and/or can retain *TNF* mRNA at the translation machinery. Irrespective of the mechanism, our findings support previous reports suggesting a functional decoupling (Kwak et al, 2022; Crawford et al, 1997), and demonstrate that the polyA length is not a key predictor for mRNA abundance and translation of TNF in the context of immune activation.

Interestingly, the effects of HuR deficiency on *IFNG* mRNA substantially differ from those of *TNF* mRNA. HuR deficiency resulted in increased transcription of *IFNG* even in the absence of T cell activation, suggesting that HuR suppresses undesired *IFNG* mRNA transcription in unstimulated T cells. During the first hour of activation, HuR-KO Teff cells maintain elevated levels of both nascent and mature *IFNG* mRNA. This, however, does not result in efficient translation into protein. Intriguingly, *IFNG* mRNA does not depend on HuR for splicing or polyadenylation. It is conceivable that HuR could regulate the nuclear export of *IFNG* mRNA during early T cell activation, as previously described for other target mRNAs (Cherry et al, 2008; Zhang et al, 2016). Additionally, the subsequent decline in *IFNG* transcription at 2 h may reflect a feedback mechanism, where the accumulated mRNA in the nucleus at 1 h (Fig. 5B) triggers transcription repression, as reported in previous studies (Berry et al, 2022). Irrespective of the exact mode of action on *IFNG* mRNA, our data highlight that HuR's activity is target-specific, as was previously reported for other RBPs (Zandhuis et al, 2021; Van Nostrand et al, 2020).

In summary, T-cell smFISH is a sensitive and quantitative tool with single-cell, single-molecule resolution for deciphering post transcriptional regulation in T cells. We found that differential mechanisms govern *IFNG* and *TNF* mRNA kinetics and showcased the contribution of HuR in regulating the cytokine production of T cells in a transcript-specific manner. We propose that T-cell smFISH is a powerful tool to uncover molecular mechanisms underlying dysregulated cytokine production, such as impaired production upon chronic stimulation (Wherry et al, 2007) or excessive production in immunopathology (Hu and Ivashkiv, 2009; Jang et al, 2021; Karki et al, 2021). The custom-made T-cell

smFISH pipeline we present here could support further studies on RNA expression and RBP-mediated regulation that define T cell function.

# Methods

### Reagents and tools table

| Reagent/resource | Reference or source | Identifier or catalog number |
|---|---|---|
| **Experimental models** | | |
| Human T cells | PBMCs, Sanquin | N/A |
| **Recombinant DNA** | | |
| **Antibodies** | | |
| HIT3a α-CD3 | BioLegend | 300302 |
| α-CD28 (CD28.2) | BioLegend | 302902 |
| Pelicluster CD3 | Sanquin | 564364 |
| DAPI | Thermo Scientific | D1306 |
| Anti-HuR clone 3A2 | Santa Cruz Biotechnology | sc-5261 |
| Mouse IgG1 kappa isotype control (P3.6.2.8.1) | eBioscience | 14-4714-82 |
| Anti-mouse-HRP | Southern Biotech | 1031-05 |
| Anti RhoGDI | Abnova | MAB9959 |
| Goat anti-rabbit-HRP | Southern Biotech | 4050-05 |
| Goat anti-mouse-HRP | Southern Biotech | 1031-05 |
| BUV737 mouse anti-human CD4 (clone SK3) | BD Bioscience | 564305 |
| BUV805 mouse anti-human CD8 (clone SK1) | BD Bioscience | 612889 |
| BV786 mouse anti-human TNF (clone MAb11) | BD Bioscience | 569461 |
| AF647 mouse anti-puromycin (clone 12D10) | Merk | MABE343-AF647 |
| AF488 mouse anti-puromycin (clone 12D10) | Merk | MABE343-AF488 |
| **Oligonucleotides and other sequence-based reagents** | | |
| *IFNG* probes smFISH | Nicolet et al, 2017 | N/A |

| Reagent/resource | Reference or source | Identifier or catalog number |
|---|---|---|
| *TNF* probes smFISH | Nicolet et al, 2017 | N/A |
| HuR sgRNA 5'-TGTGAACTACGTGACCGCGA-3' | Popović et al, 2023 | N/A |
| Non-targeting negative control crRNA #1 | IDT | 1072544 |
| IFNg RT-PCR primers | This study | N/A |
| TNF RT-PCR primers | This study | N/A |
| RT-qPCR IFNg primers | Popović et al, 2023 | N/A |
| RT-qPCR TNF primers | Popović et al, 2023 | N/A |
| RT-qPCR IL2 primers | Popović et al, 2023 | N/A |
| PAT-anchor | Meijer et al, 2007 | N/A |
| PAT-R1 | Meijer et al, 2007 | N/A |
| PAT IFNg 3'UTR F | This study | N/A |
| PAT TNF 3'UTR F | This study | N/A |
| **Chemicals, enzymes and other reagents** | | |
| Lymphoprep density gradient separation | Stemcell Technologies | # 18060 |
| CD8⁺ T cell isolation kit | Miltenyi Biotec | 130-096-495 |
| Human Serum | Sanquin | N/A |
| GolgiPlug Protein Transport Inhibitor (containing Brefeldin A) | BD Biosciences | BD 555029 |
| Recombinant human IL-15 | Peprotech | 200-15 |
| Recombinant human IL2 | Peprotech | 200-02 |
| Actinomycin D | Sigma-Aldrich | A9415 |
| Alt-R™ S.p. Cas9 Nuclease V3 | IDT | 10000735 |
| HaltTM Protease and Phosphatase Inhibitor Single-Use Cocktail, EDTA-Free (100X) | Thermo Scientific | 78443 |
| Nuclease-Free Duplex buffer | IDT | 11-05-01-12 |
| P2 primary nucleofector buffer | Lonza | V4XP-2024 |
| Puromycin dihydrochloride | Sigma | P8833-10MG |
| Alcian blue in 3% acetic acid | Sigma | B8438 |
| 32% solution PFA, EM grade | Electron Microscopy Science | 15714 |
| Formamide, Deionized | Millipore | S4117 |
| RNase-Inhibitor, Murine | New England Biolabs | M0314S |
| Yeast tRNA | Invitrogen | 15401029 |
| UltraPure Salmon Sperm DNA solution | Invitrogen | 15632011 |
| Dextran Sodium Sulfate | Sigma | 42867 |
| UltraPure BSA | Invitrogen | AM2616 |
| DAPI | Thermo Scientific | D1306 |
| Prolong Diamond Antifade Mountant | Invitrogen | P36965 |

| Reagent/resource | Reference or source | Identifier or catalog number |
|---|---|---|
| Quick-RNA Miniprep Kit | Zymo Research | R1055 |
| Maxima First Strand cDNA Synthesis Kit | Thermo Fisher Scientific | K1642 |
| Power SYBR Green | Applied Biosystems | 4367659 |
| GoTaq G2 Flexi polymerase | Promega | M7805 |
| RNA ligase 2, truncated KQ | NEB | M0373 |
| SuperScript III Reverse Transcriptase | Invitrogen | 18080044 |
| GoTaq G2 Flexi Polymerase | Promega | M7805 |
| 1Mm DDT | Invitrogen | P232 |
| RNase OUT | Invitrogen | 10777019 |
| Ribonucleoside Vanadyl Complex | NEB | S1402S |
| Protein G Dynabeads | Invitrogen | 10003D |
| RIPA lysis buffer | Thermo Fisher Scientific | 89900 |
| Near-IR | Life Technologies | L10119 |
| Cytofix/Cytoperm kit | BD Biosciences | BDB554722 |
| eBioscience™ Foxp3/ Transcription Factor Staining Buffer Set | Invitrogen | 00-5523-00 |
| **Software** | | |
| Stellaris Probe Designer tool | LGC Biosearch Technologies | N/A |
| CellSens Software | Olympus | N/A |
| FISH-quant Matlab | Mueller et al, 2013 | N/A |
| CellPose v2.2 | Stringer et al, 2020 | N/A |
| Primer3Plus | Rozen et al, 2000 | N/A |
| Fiji version 2.15.1 | ImageJ | N/A |
| FlowJo version 10.8.1 | BD Bioscience | N/A |
| RStudio environment version 4.1.1 | RStudio | N/A |
| GraphPad Prism version 9.1.1 | Prism | N/A |
| **Other** | | |
| 16-mm Fisherbrand Borosilicate Glass Circle Coverslip | Fisherbrand | 12323148 |
| Olympus BX-63 epifluorescent microscope | Olympus | N/A |
| StepOne Plus | Applied Biosystems | N.A |
| iBlot | Thermo Fisher Scientific | N/A |
| FACS Symphony A5 Cell Analyze | BD Bioscience | N/A |

## T cell activation and cell culture

Human T cells from anonymized healthy donors were used in accordance with the Declaration of Helsinki (Seventh Revision, 2013)

after written informed consent (Sanquin). Peripheral blood mononuclear cells (PBMCs) were isolated by Lymphoprep density gradient separation (Stemcell Technologies). To generate Teff cells, CD8[+] T cells were enriched from cryopreserved, defrosted PBMCs with the CD8[+] T cell isolation kit (Miltenyi Biotec) with a purity of >85%. T cells were activated for 72 h with 1 µg/mL plate-bound α-CD3 (HIT3a) and 1 µg/mL soluble α-CD28 (CD28.2; Biolegend), as previously described (Popović et al, 2023). Cells were cultured at 37 °C, 5% CO$_2$ in culture medium (IMDM, GIBCO, Thermo Fisher Scientific, supplemented with 10% fetal bovine serum (FBS), 100 U/mL penicillin, 100 µg/mL streptomycin, and 2 mM L-glutamine). Cells were harvested and cultured at a density of $1.5 \times 10^6$/mL for 7 days in standing T25/75 tissue culture flasks (Thermo Scientific) in culture medium supplemented with 100 IU/mL recombinant human (rh) IL2 (Proleukin). Medium was refreshed every 2 days. Upon nucleofection, T cells were cultured in T cell mixed media (Miltenyi Biotec) supplemented with 5% heat-inactivated human serum, 5% FBS, 100 U/mL Penicillin, 100 µg/mL streptomycin, 2 mM L-glutamine, 100 IU/mL rhIL-2.

## Gene-editing of primary human CD8[+] T cells

Cas9 RNP production and nucleofection was performed as previously described (Popović et al, 2023). sgRNA targeting HuR (5′-TGTGAACTACGTGACCGCGA-3′ (Popović et al, 2023) was dissolved in Nuclease Free Duplex buffer (Integrated DNA Technologies, IDT). Non-targeting negative control crRNA #1 (IDT) was mixed with tracrRNA at equimolar ratios (100 uM) in nuclease-free PCR tubes and denatured at 95 °C for 5 min. Nucleic acids were cooled down to RT prior to mixing and incubating for at least 10 min at RT with 30 µg Alt-R™ S.p. Cas9 Nuclease V3 (IDT). About $3 \times 10^6$ activated T cells (72 h) were nucleofected with the generated Cas9 ribonuclear proteins (RNPs) in 20 µl P2 buffer (Lonza) in 16-well strips using program EH100 in a 4D Nucleofector X unit (Lonza). HuR depletion was confirmed on day 5 after nucleofection by flow cytometry, as described below.

## T cell activation

Teff cells were stimulated with 1 µg/ml soluble α-CD3 (Pelicluster CD3, Sanquin) and 1 µg/mL soluble α-CD28 in culture media for the indicated time points. For intracellular cytokine staining measurements, 1 µg/mL brefeldin A (BD Bioscience) was added during the last 2 h of activation. For translation efficiency, T cells were incubated with Puromycin dihydrochloride (Sigma) for 10 min at 37 °C. For mRNA stability measurements, T cells were activated for 1 h with α-CD3/ α-CD28 and then treated for 1 h with 5 mg/mL actinomycin D (ActD) (Sigma-Aldrich) and/or with 5 µg/mL of Harringtonine (Abcam).

## smFISH probe design

Probes for *IFN* and *TNF* were previously described (Nicolet et al, 2017), and sequences are provided in Appendix Table S1. Briefly, probes were designed using the Stellaris Probe Designer tool (from LGC Biosearch Technologies). Subsequent probe blasting resulted in the removal of a couple of probes that showed predicted high affinity for secondary target genes. This resulted in 37 and 45 probes for *IFNG* and *TNF*, respectively. Because we observed

autofluorescent granule structures in activated T cells in the FITC and Cy3 channels, CALfluorRed 610 and Quasar 670 were used as conjugates for IFNG and TNF probes, respectively (Appendix Methods). Probes were purchased from LGC BioSearch Technologies.

## Two-color single-molecule fluorescent in situ hybridization (smFISH)

Two-color smFISH was performed as described in (Maekiniemi et al, 2020; Vera et al, 2019). Optimization of T cell attachment, cell amount and fluorophore selection are described in Appendix Methods. RNase-free reagents were used throughout.

Coverslips coating and seeding:

1) Round 16-mm coverslips (Fisherbrand Borosilicate Glass Circle Coverslip) were cleaned with 0.5 M HCl, washed twice with Milli-Q water,
2) Coverslips were coated with Alcian blue (Alcian blue in 3% acetic acid, Sigma) for 30 min at RT and washed (x6 times minimum) with Milli-Q water
3) $1 \times 10^6$ Teff cells were seeded in 60 µl ice-cold 1xPBS per coverslip to prevent autolysis and stop any ongoing cellular process. T cells were incubated for 20 min at 37 °C to enhance T cell attachment.
4) Coverslips were gently washed once with ice-cold PBS to remove non-attached cells.
5) Teff cells were fixed with 4% paraformaldehyde (32% solution, EM grade; Electron Microscopy Science #15714) for 10 min at RT in the dark.
6) Cells were washed twice with quenching buffer (3 mM MgCl$_2$, 0.1 M glycine in 1xPBS) for 5 min at RT. Coverslips were gently covered with 70% ice-cold ethanol and stored at −20 °C until further use, both to decrease autofluorescence, and to preserve RNA and cellular structure.

Coverslips pre-hybridization:

1) Coverslips were rehydrated by incubating twice with 1 ml 2xSSC for 5 min.
2) Coverslips were incubated with pre-hybridization buffer (2xSSC, 10% deionized formamide, 1:1000 RNase-inhibitor in ultrapure distilled water) for 50 min at RT.

Probes preparation and hybridization:

1) Hybridization solution A was prepared with 20% formamide to denature single-strand DNA, supplemented with 10 µg/ml yeast tRNA (Invitrogen) and 10 µg/ml salmon sperm DNA (Ultrapure salmon sperm DNA solution, Invitrogen), used as nucleic acid competitors to saturate nonspecific probe binding.
2) IFNγ-CF610 and TNF-Quasar 670 probes were freshly diluted to a final concentration of 125 nM in hybridization solution A.
3) Hybridization solution A was denatured for 5 min at 95 °C.
4) Hybridization solution B was prepared as follows. Hybridization solution B includes reagents that are temperature sensitive. Specifically, we use 10% dextran sodium sulfate to enhance probe hybridization, 10 mg/ml BSA to reduce background signal, 1:1000 murine RNase-inhibitor to preserve RNA and 2xSSC in ultrapure water.

5) Hybridization solution A was added to 1:1 ratio with Hybridization solution B.

6) Hybridization was performed for 3 h at 37 °C in the dark.

7) Coverslips were washed twice with pre-hybridization buffer for 15 min at 37 °C, followed by two washes with 2xSSC for 10 min at RT.

Nuclear staining and coverslip mounting:

8) Coverslips were incubated with 1xSSC for 5 min at RT

9) Coverslips were incubated with 1 µg/ml DAPI (Thermo Scientific) for 5 min in the dark at RT.

10) Coverslips were washed twice at RT with 1xPBS and mounted on glass with Prolong Diamond Antifade mountant (Invitrogen).

## T cell smFISH image acquisition and analysis

Images were acquired as previously described (van Otterdijk et al, 2024). Briefly, image acquisition was performed using an Olympus BX-63 epifluorescence microscope equipped with an Ultrasonic stage and UPLSAPO 100 × 1.4NA oil immersion objective (Olympus). Lumencore SOLA FISH light source, Hamamatsu ORCAFusion sCMOS camera (6.5-µm pixel size) mounted using U-CMT C-Mount Adapter, and zero-pixel shift filter sets: F36-500 DAPI HC Brightline Bandpass Filter, F36-502, FITC HC BrightLine Filter, F36-542 Cy3 HC BrightLine Filter, AHF-LED-FISH-R Filter for Cy3.5 and F36-523 Cy5 HC BrightLine filter. Z-section of 200 nm intervals over an optical range of 12 µm. Per each coverlip, 12 positions of 61 Z-stack were collected to obtain measurements from at least 500 cells per each donor per time point. The CellSens software (Olympus) is used for instrument control and image acquisition.

smFISH images were analysed with FISH-quant Matlab (Mueller et al, 2013). FISH-quant allows to quantify both nascent and mature RNAs. Briefly, the probes designed to detect nascent and mature RNAs are the same set of 40–50 oligonucleotides, that target the coding sequence. The distinction between nascent and mature RNAs is based on their location—nuclear vs cytoplasmic—and on their intensity- nascent mRNAs are located in nuclear fluorescent clusters that have an intensity at least 1.5 times brighter than the average intensity of cytoplasmic mRNA. We opted for this approach because mammalian mRNAs are transcribed in bursts, i.e., multiple polymerases transcribe a gene during a transcription event. As a result, transcription sites are generally bright fluorescent clusters, and easily distinguishable from cytoplasmic mRNAs. After background subtraction, the Transcript Site Outline tool (FISH-quant) was used to identify transcription sites as high-intensity signals (x1.5 intensity of averaged mRNA) colocalizing with DAPI staining. Mature RNA quantification was performed by fitting RNA spots to a three-dimensional (3D) Gaussian. The intensity and width of the 3D Gaussian tool of FISH-quant were thresholded to exclude nonspecific signals. To count nascent RNAs at a transcription site, the FISH-quant pipeline includes a step where the average intensity and size (i.e., 3D sphere) of all cytoplasmic mRNAs is computed. This information is then used to estimate the number of nascent RNAs at the transcription site. Of note, quantification of nascent RNA molecules is based on the intensity and size of full-length mRNAs in the cytoplasm. Because nascent RNA at the transcription site could also be partial RNAs in the process of being synthesized, using one set of probes to detect both

nascent and mature RNAs may slightly underestimate the number of nascent RNAs at the transcription sites. Therefore, a second probe set for intergenic sequences would be useful for genes that are transcribed in small bursts (1–2 nascent RNAs). This is, however, not the case for cytokines, which display substantial induction of transcriptional activity.

Data were post-processed using Filtering.Rmd script (https://github.com/nikolinasostaric/T-cell_smFISH). With the 2D-outline output from FISH-quant, cells in division (with two DAPI-stained nuclei), cells on the edge of the coverslips, and cells with miscalled nuclei were excluded from further analysis (~20% of the cells/experiment).

Subcellular localization of RNA was defined with the 3D spot localization pipeline. 3D segmentation of nuclei was performed with CellPose v2.2 (https://www.cellpose.org/) (Stringer et al, 2020). DAPI staining over *z*-stacks was used for the nuclear outline. The following settings were employed: cytoplasm model 2.0 (cyto2), 3D setting stitch_threshold >0, and nuclei diameter were calculated as the average diameter of images per each time point of activation. Nuclear mask coordinates and coordinates (*x,y,z*) per each RNA spot were imported into Spot_localization.ipynb script. *x,y*, and *z* coordinates were used to define the colocalization of each mRNA with the nuclear mask (per respective image). mRNA spots with coordinates not matching the nuclear mask were considered cytoplasmic mRNA.

## Quantitative (RT-qPCR) and reverse transcription-PCR (RT-PCR) analysis

Total RNA was extracted from T cells using the Quick-RNA Miniprep Kit (Zymo Research, R1055) according to the manufacturer's protocol. cDNA was generated with Maxima First Strand cDNA Synthesis Kit (Thermo Fisher Scientific, K1642). To study cytokine mRNA expression, RT-qPCR was performed with duplicate reactions using Power SYBR Green (Applied Biosystems, 4367659) on a StepOne Plus (Applied Biosystems). Ct values were normalized to 18S levels. *IFNG*, *TNF*, and *IL2* primers used were previously described. For splicing RT-PCR, 200 ng cDNA was amplified with GoTaq G2 Flexi polymerase (Promega, M7805) with human *IFNG* and *TNF* exon- and intron-specific primers, designed by using the Primer3Plus (Rozen and Skaletsky, 2000) (Appendix Table S2) with the following protocol: 95 °C for 1 min, 30 cycles of (30 s at 95 °C, 30 s at 50 °C, 2.5 min at 72 °C), followed by 5 min at 72 °C. PCR products were run on a 1–1.2% agarose gel. Quantification was performed with Analyze-Gel of Fiji version 2.15.1.

## RNA ligation-mediated poly(A) test (RL-PAT)

RL-PAT was performed as described (Meijer et al, 2007). The 5′ to 5′ adenylated and 3′ blocked 'PAT-anchor' oligo was ligated to the 3′ end of total RNA overnight at 16 °C with RNA ligase 2, truncated KQ (NEB, M0373). To generate cDNA, ligated RNA was reverse transcribed with SuperScript III Reverse Transcriptase (Invitrogen, 18080044) with the "PAT-R1" oligo (complementary to "PAT-anchor"). cDNA was then amplified with GoTaq G2 Flexi Polymerase (Promega, M7805), using a forward primer annealing to the 3′ UTR of the mRNA of interest (Appendix Table S3) and PAT-R1 as the reverse primer. All mRNA-specific PAT primers

were validated by performing PAT on mRNA that was deadenylated with oligo-d(T) and treated with RNAse H.

## RNA immunoprecipitation and immunoblotting

RNA immunoprecipitation was performed as previously described (Popović et al, 2023). Briefly, cytoplasmic lysates of $20 \times 10^6$ human CD8[+] Teff cells were prepared with lysis buffer (10 mM HEPES, pH 7.0, 100 mM KCl, 5 mM MgCl$_2$, 0.5% NP40) freshly supplemented with 1 Mm DTT, 100 U/ml RNase OUT (both Invitrogen), 0.4 Mm Ribonucleoside Vanadyl Complex (NEB), and 1% EDTA-free protease/phosphatase inhibitor cocktail (Thermo Scientific). Protein G Dynabeads (Invitrogen) were prepared according to the manufacturer's protocol. The lysate was immunoprecipitated for 4 h at 4 °C with 10 mg mouse monoclonal a-HuR (3A2, Santa Cruz Biotechnology) or a mouse IgG1 kappa isotype control (P3.6.2.8.1, eBioscience). RNA was extracted from beads with Trizol, and mRNA expression was measured by RT-PCR. The specificity of the RNA-IP was validated by immunoblotting a-HuR, followed by goat anti-mouse-HRP (1031-05, Southern Biotech). Cell lysates ($1 \times 10^6$ cells/sample) were prepared using standard procedures with RIPA lysis buffer. Proteins were separated on a 4–12% SDS/PAGE and transferred onto a nitrocellulose membrane by iBlot (Thermo). Mouse monoclonal a-HuR and anti-RhoGDI (MAB9959, Abnova), were used, followed by goat a-rabbit (4050-05) and goat a-mouse-HRP secondary antibodies, respectively (1031-05, both Southern Biotech).

## Flow cytometry and intracellular staining

T cells were washed with FACS buffer (PBS with 1% FBS and 2 mM EDTA) and labeled for 20 min at 4 °C with α-CD4 (SK3, BD Horizon), α-CD8 (SK1, BD Horizon). Dead cells were excluded with Near-IR (Life Technologies). For intracellular staining, cells were fixed and permeabilized with Cytofix/Cytoperm kit (BD Biosciences), and stained with α-IFN-γ (4S.B3, BD Bioscience), α-TNF (MAb11, BD Bioscience), α-Puromycin (12D10, Merk). Acquisition was performed using the FACS Symphony A5 Cell Analyzer (BD Bioscience). For HuR staining, cells were fixed and permeabilized with eBioscience™ Foxp3/Transcription Factor Staining Buffer Set (Invitrogen) prior to staining with α-HuR (3A2, Santa Cruz Biotechnology), according to the manufacturer's protocol. Data were analyzed with FlowJo (BD Biosciences, version 10.8.1).

## Data analysis of flow cytometry data

The FlowJo workspace (.xml or .wsp file) and its FCS files were imported into the R environment (version 4.1.1) (Foundation, R. R: The R Project for Statistical Computing. https://www.r-project.org/), creating a GatingSet object using CytoML (version 2.4.0) (Finak et al, 2018). Raw intensity data from gated flow data (IFN-γ, or TNF single positive, and IFN-γ/TNF double positive, or double negative) were extracted from GatingSet using flowWorkspace (version 4.4.0) (Finak and Jiang, 2018) and flowCore (version 2.4.0) (Hahne et al, 2009).

## Data visualization

Results are shown as mean ± SD. Statistical analysis was performed in R-studio, with a two-tailed ratio paired or unpaired Student's *t*-test when comparing two groups, or with Kruskal–Wallis non-

parametric test with Tukey-HSD correction when comparing multiple groups over different time points. *p* values <0.05 were considered statistically significant. Data were visualized with ggplot2 (version 3.4.2) (Wickham, 2011) or using GraphPad Prism (version 9.1.1).

## Data availability

Code and scripts have been deposited on GitHub (https://github.com/nikolinasostaric/T-cell_smFISH). Microscopy images have been deposited on the BioImages Archive, accession number S-BIAD2255.

The source data of this paper are collected in the following database record: biostudies:S-SCDT-10_1038-S44318-025-00592-0.

## Peer review information

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

## Acknowledgements

We thank K. Bresser, A.P. Jurgens, and N.D. Zandhuis for critical reading of this manuscript. This research was supported by the European research council ERC-Printers 817533 (MCW), and by Oncode Institute (MCW).

## Author contributions

**M Valeria Lattanzio**: Conceptualization; Resources; Data curation; Software; Formal analysis; Validation; Investigation; Visualization; Methodology; Writing —original draft; Project administration; Writing—review and editing. **Nikolina Šoštarić**: Data curation; Software; Formal analysis; Visualization. **Nandhini Kanagasabesan**: Data curation; Software; Formal analysis; Visualization. **Branka Popović**: Investigation; Methodology. **Antonia Bradarić**: Validation; Methodology. **Leyma Wardak**: Validation; Methodology. **Aurélie Guislain**: Validation; Investigation; Methodology. **Philipp Savakis**: Data curation; Software; Formal analysis; Visualization. **Evelina Tutucci**: Conceptualization; Writing—original draft; Project administration; Writing—review and editing. **Monika C Wolkers**: Conceptualization; Supervision; Funding acquisition; Writing—original draft; Project administration; Writing—review and editing.

Source data underlying figure panels in this paper may have individual authorship assigned. Where available, figure panel/source data authorship is listed in the following database record: biostudies:S-SCDT-10_1038-S44318-025-00592-0.

## Disclosure and competing interests statement

The authors declare no competing interests.

