## [Peer Review File · The EMBO Journal]

Single-molecule imaging of transcription dynamics, RNA localization and fate in human T cells

M. Valeria Lattanzio, Nikolina Šoštarić, Nandhini Kanagasabesan, Branka Popović, Antonia Bradarić, Leyma Wardak, Aurélie Guislain, Philipp Savakis, Evelina Tutucci, and Monika C. Wolkers

Corresponding author(s): *Evelina Tutucci (evelina.tutucci@vu.nl) and Monika C. Wolkers (m.wolkers@sanquin.nl)*

Review Timeline:

Transfer from Review Commons:	14th Jul 25
Editorial Decision:	8th Aug 25
Revision Received:	2nd Sep 25
Accepted:	26th Sep 25

Review
COMMONS

Editor: Hartmut Vodermaier

Transaction Report: This manuscript was transferred to The EMBO JOURNAL following peer review at Review Commons.

Review #1

1. Evidence, reproducibility and clarity:

Evidence, reproducibility and clarity (Required)

The manuscript by Lattanzio and colleagues uses advanced single molecule FISH, adapted specifically for T cells, to examine RNA transcriptional dynamics in polyclonally stimulated human T cells. By examining the subcellular localisation of both IFN γ and TNF mRNAs (nascent and mature), they are able to characterise rate things like rate of transcription and RNA stability. Key findings include the identification of bi-allelic vs mono-allelic transcription at the single cell level which maps to polyfunction vs monofunctional T cells. Moreover, they identified distinct mechanisms regulating RNA stability and the role of RNA binding protein HuR in mediating that.

Overall, this is really a proof of concept paper that uses elegant technologies and analysis tools showing just how much information can be obtained from this approach. The ability to examine RNA dynamics and the impact of RNA binding proteins in regulating RNA stability/translation/transcriptin at a single cell level will be an advance for the field, not just those interested in T cell biology but all cell types.

There are no specific experimental issues that came to mind that need to be addressed and it is really only some minor comments, particularly for the discussion that would strengthen the implications of the study.

1. I might have missed it but it wasn't exactly clear from the results or from the methods exactly how nascent vs mature RNA was discriminated. Was this just from the subcellular localisation (i.e nuclear vs cytoplasmic)? RNA imaged close to the TSS? If so, this should be noted somewhere. If there was some other way of precisely ascribing RNA status, this should be outlined (use of primers that targeted intergenic sequences).

2. The discussion was very brief and would have benefited from a bit more speculation about the implications of their findings. Specifically, why would there be a need for different cytokine RNAs to be regulated in such distinct ways (IFN γ vs IL-2)? Do the authors have any thoughts? Another point is the proposed explanations for the distinct T cell subsets observed that produce cytokines at different levels. While the authors propose three possible explanations, they are presented as being mutually exclusive, of course it could be a combination of all three. Moreover, putting the importance of higher functioning T cells (i.e those that produce more cytokines) into context is also important. Early studies from La Gruta et al., (J Immunol, 2004; doi: 10.4049/jimmunol.172.9.5553), Betts et al,

(Blood, 2006 doi.org/10.1182/blood-2005-12-4818) and Darrah et al. (Nat Med, 2007 doi.org/10.1038/nm1592) linked higher cytokine production/multifunctionality to better immune outcomes, while Denton et al (PNAS, 2011, 10.1073/pnas.1112520108) linked the extent of cytokine production to cellular differentiation (and epigenetic landscape at the TNF and IFNg locus). These studies should be cited to provide a setting where this approach will be relevant.

A minor point relates to line 329, that sentence stating "Even though most activated Teff cells express cytokine mRNAs, they display a two order of magnitude difference in mRNA and protein expression."

It is not clear what this is relevant or compared to. A two order of magnitude difference compared to what?

2. Significance:

Significance (Required)

This is a proof of concept study that demonstrates the utility of the T cell smFISH approach to delineate high resolution analysis of cytokine RNA dynamics at a single cell level, for multiple cytokine RNA species. It clearly provides interesting biology and further understanding of RNA dynamics in activated T cells. I especially appreciated the observation of bi-allelic vs mono-allelic transcription, and the ability to explore the role of RNA binding proteins in RNA regulation.

This technique will have broader applicability and hence will be of interest to those outside T cell immunology. It only requires some minor corrections/revisions.

3. How much time do you estimate the authors will need to complete the suggested revisions:

Estimated time to Complete Revisions (Required)

(Decision Recommendation)

Less than 1 month

4. Review Commons values the work of reviewers and encourages them to get credit for their work. Select 'Yes' below to register your reviewing activity at Web of Science Reviewer Recognition Service (formerly Publons); note that the content of your review will not be visible on Web of Science.

Yes

Review #2

1. Evidence, reproducibility and clarity:

Evidence, reproducibility and clarity (Required)

The authors describe a modified version of single molecule Fluorescence In-situ Hybridization (smFISH) method they have adapted to successfully measure RNA levels in isolated human donor T cells, that are very hard to grow on glass and have small amounts of cytoplasm relative to cell size, a challenge for all researchers working with small cells that only grow in suspension cultures. Using this methodology, the authors have queried transcription status and mRNA localization and fate of the two cytokines, IFNG and TNF, upon T-cell activation. The main findings of the study are: (1) activation of T-cells results in rapid accumulation of IFNG and TNF mRNA; there is differential distribution of the cytokine mRNAs between the nucleus and cytoplasm with greater accumulation in the cytoplasm as activation progresses resulting in increased protein production. There is significant transcriptional heterogeneity in response to T-cell activation. (2) The cytokine mRNA turnover appears to be controlled by translation. (3) HUR, an RBP appears to control poly(A) tail length of TNF mRNA in response to T-cell activation.

The successful implementation of a modified smFISH protocol used in this study is a welcome resource for all labs that want to study small human primary cells that are difficult to culture on glass coverslips and grow as suspension cultures.

Although the authors have very exciting observations, they have shied away from discussing their results in the context of the biology of T-cell activation and how their observations may explain prior studies on cytokine gene expression patterns during T-cell activation.

In my opinion, the authors should discuss their observations in depth from the context of T-cell activation and cytokine expression. I have enumerated several specific comments that may help the authors in revising the manuscript if they choose to do so.

****Specific comments:****

1. Based on the data presented in Figure 1 D and E, it is clear there is depletion of IFNG and

TNF mRNA 4hrs after activation and then the mRNA levels go up at 6h in both cases. However, the authors suggest that only TNF mRNA is depleted at 4hrs of activation (lines 169-172). The median number of IFNG mRNA gradually decreases after 1h of activation and reaches a low at 4h and then substantially increases by 6h. Did the authors measure gene expression of these mRNAs at later time points in the activation process? Perhaps transcription is coupled to mature mRNA levels in the cytoplasm and transcription is ramped up again once the cytoplasmic mRNA levels reach a lower threshold. Is this just an anomaly of the system or is gene expression pattern of cytokines upon T-cell activation cyclical?

2. In data presented Figure 2 and Suppl Figure 2, the authors show correlation between dual cytokine expression and biallelic expression. However, not all dual cytokine expressing cells show bi-allelic expression of both cytokines. It will be useful to know what fraction of cells are biallelic for both genes. Since the experiment was done using two color smFISH, a scatter plot will cluster those dual expressor cells for both cytokines that are also bi-allelic for both genes. Extending this further would be to systematically address protein expression in the various combination of expression patterns. Combining smFISH with immunofluorescence will help address this. Overall, these results will be helpful in getting a better understanding of gene expression patterns during T-cell activation.

3. The mRNA localization data presented in Figure 3A and the associated supplemental figure: A better analysis and representation of the data presented in 3A would be a scatter plot of individual cells for their nuclear and cytoplasmic localization of mature mRNA. The authors might also want to extend this analysis based on the data presented in Figure 2 for dual expressors and bi-allelic expression. In other words, do cells with bi-allelic expression have more mRNA localized in the cytoplasm, and does this hold true in dual expressor cells? In the context of translation dependent decay of mRNA, do the dual expressor cells with biallelic expression fare better thereby producing and secreting cytokines continuously?

4. The data presented for IFNG in Figure 4 is quite intriguing. In HuR-KO cells at 2h post induction, two of the three donor cell lines have only a small fraction of cells producing protein compared to the controls, however, they are substantially higher than the KO cells at time "0". Surprisingly, the amount of protein produced by these cells (panel B), although statistically lower than the control, is substantially higher than KO cells at "0" h. Does the lone donor cell line with higher number of protein producing cells contribute to majority of the protein produced? There appears to be substantial difference between the three donor cell lines in the number of protein producing cells and mature IFNG mRNA after activation (Suppl Figure 1G & H). The authors may wish to compare the results before combining the data of all three donor cell lines before interpreting the data.

5. Also intriguing, HuR knock out results in a significant increase in transcription of IFNG at

time "0" (Figure 4, panel E). Despite this, there is a significant loss in transcription of IFNG 2h post activation. However, there is significant accumulation of mature mRNA (panel D). Combined with the protein expression data presented in panels A & B, and the fact that translation induces mRNA decay, how do the authors reconcile this data?

6. The differential effect of HuR knock out on poly(A) tail length of IFNG and TNF mRNA is of great importance and the most striking finding in this study! It is generally accepted that poly(A) tail length contributes to mRNA stability and survival. The results presented in Figures 4 and 5 argue otherwise. Only a small fraction of TNF mRNA have full length poly(A) tails, however, the number of mature TNF mRNA in KO cells is much greater than the control even at "0"h. In addition, the TNF mRNA appear to be well translocated into the cytoplasm and effectively translated. Given these conflicting observations, what possible mechanism do the authors envision that can explain this result.

Again, plotting the data presented in Figure 5A as a scatter plot between # of RNA in the cytosol vs nucleus will give a better picture of the localization changes in individual cells.

7. A more elaborate discussion of the results as it relates to the biology of cytokine gene expression during T-cell activation will immensely strengthen the manuscript.

****Minor comment:****

1. Images of cells with smFISH data (Figures 1, 3 & 4) must be bigger for better visualization. Show images with only a couple of cells enlarged to show the mRNA spots more visibly. Include images with more cells in the supplement instead.

****Referee's cross-commenting****

I must confess I am not an immunologist, so my knowledge of the intricacies of gene expression in T-cells is very limited. However, I do have a fair sense of transcription regulation and use single molecule approaches, especially smFISH, to address these questions. I agree with the other reviewer the study is of significance, especially the advancement in the ability to do smFISH in primary cells, a challenge that I know first hand. I also have to agree with the other reviewer that the discussion was too short and the authors shied away from the bigger picture of being able to comment on regulation of expression of cytokines during T-cell activation. It is remarkable that they see heterogeneity in gene expression of the individual target genes and bi-allelic expression. The other point of interest is the difference in p(A) tail length and its potential role in regulating TNF gene expression.

2. Significance:

Significance (Required)

The successful implementation of a modified smFISH protocol used in this study is a welcome resource for all labs that want to study small human primary cells that are difficult to culture on glass coverslips and grow as suspension cultures.

Overall, this work is of high quality and can be better presented to fully explore and discuss the biological implications of the observations from the study. It is not clear to me if the authors wished to present this manuscript reporting an advancement in technology tool to study gene expression during T-cell activation, or a more in-depth study of gene expression.

The study will benefit the larger community that use single molecule approaches to understand gene expression.

3. How much time do you estimate the authors will need to complete the suggested revisions:

Estimated time to Complete Revisions (Required)

(Decision Recommendation)

Between 1 and 3 months

Yes

Full Revision

Manuscript number: RC-2025-03025

Corresponding author(s): Monika C. Wolkers, Evelina Tutucci

[Please use this template only if the submitted manuscript should be considered by the affiliate journal as a full revision in response to the points raised by the reviewers.]

*If you wish to submit a preliminary revision with a revision plan, please use our "Revision Plan" template. **It is important to use the appropriate template to clearly inform the editors of your intentions.***

1. General Statements [optional]

This section is optional. Insert here any general statements you wish to make about the goal of the study or about the reviews.

We thank both reviewers for their enthusiastic and thoughtful evaluation. We are very pleased that the potential of our single molecule FISH approach to reveal RNA transcriptional dynamics, mRNA location, and the role of HuR at single-cell resolution in primary human T cells was well received. We agree that some technical parts would benefit from further explanation, and that the discussion of our findings could be extended. Please find below how we addressed the suggested improvements, which we believe have been good additions to your study.

This section is mandatory. Please insert a point-by-point reply describing the revisions that were already carried out and included in the transferred manuscript.

Reviewer #1 (Evidence, reproducibility and clarity (Required)):

The manuscript by Lattanzio and colleagues uses advanced single molecule FISH, adapted specifically for T cells, to examine RNA transcriptional dynamics in polyclonally stimulated human T cells. By examining the subcellular localisation of both IFN γ and TNF mRNAs (nascent and mature), they are able to characterise rate things like rate of transcription and RNA stability. Key findings include the identification of bi-allelic vs mono-allelic transcription at the single cell level which maps to polyfunction vs monofunctional T cells. Moreover, they identified distinct mechanisms

regulating RNA stability and the role of RNA binding protein HuR in mediating that. Overall, this is really a proof of concept paper that uses elegant technologies and analysis tools showing just how much information can be obtained from this approach. The ability to examine RNA dynamics and the impact of RNA binding proteins in regulating RNA stability/translation/transcriptoin at a single cell level will be an advance for the field, not just those interested in T cell biology but all cell types.

We thank the reviewer for this enthusiastic and thoughtful evaluation. We are very pleased that the potential of our single molecule FISH approach to reveal RNA transcriptional dynamics, mRNA location, and the role of HuR at single-cell resolution in primary human T cells was recognized.

There are no specific experimental issues that came to mind that need to be addressed and it is really only some minor comments, particularly for the discussion that would strengthen the implications of the study.

REVIEWER 1

1. I might have missed it but it wasn't exactly clear from the results or from the methods exactly how nascent vs mature RNA was discriminated. Was this just from the subcellular localisation (i.e nuclear vs cytoplasmic)? RNA imaged close to the TSS? If so, this should be noted somewhere. If there was some other way of precisely ascribing RNA status, this should be outlined (use of primers that targeted intergenic sequences).

In the revised manuscript, we now explained the methodology in more detail, especially for readers that are not familiar with smFISH analysis. We addressed it as follows:

The nascent mRNA analysis was performed with the smFISH analysis pipeline FISH-quant (previously described in Mueller et al.; 10.1038/nmeth.2406). To better explain how the pipeline identifies and quantifies nascent RNAs at transcription sites (TsX), we now included **NEW Supplementary Figure 1B** (see below). Additionally, we expanded the description in the Material and Methods section to explain how nascent and mature RNAs are distinguished (**line 475-497**). For both cytokines, we designed probes spanning the coding sequence. The probes that consisted of 40-50 oligonucleotides were used to detect both nascent and mature RNAs. To distinguish between those two, we used their location (nuclear vs cytoplasmic) and their intensity (nascent mRNAs are located in nuclear fluorescent clusters that have an intensity at least 1.5 times brighter than the average intensity of cytoplasmic mRNA). We opted for this approach because mammalian mRNAs are transcribed in bursts, i.e. multiple

polymerases transcribe a gene during a transcription event ([10.1016/j.cub.2006.03.092](https://doi.org/10.1016/j.cub.2006.03.092), [10.1038/s41556-024-01486-9](https://doi.org/10.1038/s41556-024-01486-9), [10.1371/journal.pbio.0040309](https://doi.org/10.1371/journal.pbio.0040309), [10.1073/pnas.121353010](https://doi.org/10.1073/pnas.121353010)). As a result, transcription sites are generally bright fluorescent clusters, and easily distinguishable from cytoplasmic mRNAs.

Furthermore, to count nascent RNAs at a transcription site, the FISH-quant pipeline includes a step where the average intensity and size (i.e. 3D sphere) of all cytoplasmic mRNAs is computed. This information is then used to estimate the number of nascent RNAs at the transcription site. Of note, quantification of nascent RNA molecules is based on the intensity and size of full-length mRNAs in the cytoplasm. Because nascent RNA at the transcription site could also be partial RNAs in the process of being synthesized, using one set of probes to detect both nascent and mature RNAs may slightly underestimate the number of nascent RNAs at the transcription sites. This, however, may also be true when using two sets of probes (exonic and intronic), because at least 25 probes need to bind to reach the detection limit to measure mRNA. Therefore, for the purpose of our current study we believe that the approach taken is sufficiently accurate. To highlight this point, we expanded the discussion (**lines 488-497**) and discussed the alternative validation approach using probes that target intergenic sequences, which could in particular be useful for genes that are transcribed in small bursts (1-2 nascent RNAs), cases where the identification of transcription sites based on location and intensity might not be that obvious. This is, however, not the case for cytokines, which substantially increase their transcription activity upon T cell activation.

NEW Suppl. Fig 1B. Scheme of nascent RNA quantification using FISH-quant. (1) TsXs are identified as high intensity fluorescent clusters overlapping with nuclear staining (DAPI). Their XYZ position and intensity (amplitude) are measured. (2) Cytoplasmic mRNAs are identified, their XYZ position and intensity is measured. (3) Quantification of average mRNA intensity and 3D size (amplitude and sigma XY, sigma Z) based on all mRNA measured at a given time point. (4) Estimation of the number of nascent RNA per transcription sites based on the amplitude of TsX and average mRNA.

2. The discussion was very brief and would have benefited from a bit more speculation about the implications of their findings. Specifically, why would there be a need for different cytokine RNAs to be regulated in such distinct ways (IFN γ vs IL-2)? Do the

authors have any thoughts? Another point is the proposed explanations for the distinct T cell subsets observed that produce cytokines at different levels. While the authors propose three possible explanations, they are presented as being mutually exclusive, of course it could be a combination of all three. Moreover, putting the importance of higher functioning T cells (i.e those that produce more cytokines) into context is also important. Early studies from La Gruta et al., (J Immunol, 2004; doi: 10.4049/jimmunol.172.9.5553), Betts et al, (Blood, 2006 doi.org/10.1182/blood-2005-12-4818) and Darrah et al. (Nat Med, 2007 doi.org/10.1038/nm1592) linked higher cytokine production/multifunctionality to better immune outcomes, while Denton et al (PNAS, 2011, 10.1073/pnas.1112520108) linked the extent of cytokine production to cellular differentiation (and epigenetic landscape at the TNF and IFNg locus). These studies should be cited to provide a setting where this approach will be relevant.

We thank this reviewer for these suggestions, which we have incorporated in the discussion as follows:

1) We expanded the discussion on the functional relevance of *IFNG*, *TNF* and *IL2* mRNA and protein kinetics in Teff cells (**lines 313-319**). We and others have previously reported on the differential expression kinetics of these cytokines and how (post)-transcriptional events govern their regulation. Notably, the timing of their production aligns with their specific roles in T cell responses: *IFNG* is critical for antiviral and antitumor activity, while *TNF* contributes to cytotoxicity and recruits other immune cells, which should both occur rapidly. However, premature or excessive exposure to *IFNG* can block cell division of effector T cells, and of *TNF* of unwanted cell death, both leading to immunopathology. In contrast, *IL2* production is crucial for cell proliferation and survival, but these signals can impede effector function during the first encounter with target cells. Thus, for appropriate T cell function, a timely and tightly controlled production of these cytokines is key.

2) We agree that the proposed reasons of high divergent levels of cytokine production may not be mutually exclusive, and we included this consideration (**lines 330-332**)

3) We now also better highlighted the relevance that T cells produce more than one cytokine we have now included the suggested literature (**lines 327-330**).

A minor point relates to line 329, that sentence stating "Even though most activated Teff cells express cytokine mRNAs, they display a two order of magnitude difference in mRNA and protein expression." It is not clear what this is relevant or compared to. A two order of magnitude difference compared to what?

We realize that this statement was not well phrased. We aimed to highlight that the single cell resolution of smFISH revealed a high heterogeneity of mRNA expression per cell within one T cell population. To clarify this point, we changed the wording as follows (**lines 321-324**):

*'Even though the majority of activated Teff cells expressed cytokine mRNA, the single cell resolution of smFISH revealed a wide range of mRNA expression (i.e. from 1 mRNA molecule to hundreds of mRNA molecules per cell; **Fig. 1E**). This heterogeneity within the T cell population was observed at all time points and was also observed for the protein expression (**Fig. 1E, G**).'*

Reviewer #1 (Significance (Required))

This is a proof of concept study that demonstrates the utility of the T cell smFISH approach to delineate high resolution analysis of cytokine RNA dynamics at a single cell level, for multiple cytokine RNA species. It clearly provides interesting biology and further understanding of RNA dynamics in activated T cells. I especially appreciated the observation of bi-allelic vs mono-allelic transcription, and the ability to explore the role of RNA binding proteins in RNA regulation.

This technique will have broader applicability and hence will be of interest to those outside T cell immunology. It only requires some minor corrections/revisions.

We thank this reviewer for the enthusiastic assessment of our work. We have revised the manuscript as indicated above.

REVIEWER #2 (Evidence, reproducibility and clarity (Required)):

The authors describe a modified version of single molecule Fluorescence In-situ Hybridization (smFISH) method they have adapted to successfully measure RNA levels in isolated human donor T cells, that are very hard to grow on glass and have small amounts of cytoplasm relative to cell size, a challenge for all researchers working with small cells that only grow in suspension cultures. Using this methodology, the authors

have queried transcription status and mRNA localization and fate of the two cytokines, IFNG and TNF, upon T-cell activation. The main findings of the study are: (1) activation of T-cells results in rapid accumulation of IFNG and TNF mRNA; there is differential distribution of the cytokine mRNAs between the nucleus and cytoplasm with greater accumulation in the cytoplasm as activation progresses resulting in increased protein production. There is significant transcriptional heterogeneity in response to T-cell activation. (2) The cytokine mRNA turnover appears to be controlled by translation. (3) HUR, an RBP appears to control poly(A) tail length of TNF mRNA in response to T-cell activation.

The successful implementation of a modified smFISH protocol used in this study is a welcome resource for all labs that want to study small human primary cells that are difficult to culture on glass coverslips and grow as suspension cultures. Although the authors have very exciting observations, they have shied away from discussing their results in the context of the biology of T-cell activation and how their observations may explain prior studies on cytokine gene expression patterns during T-cell activation.

In my opinion, the authors should discuss their observations in depth from the context of T-cell activation and cytokine expression. I have enumerated several specific comments that may help the authors in revising the manuscript if they choose to do so.

We thank this reviewer for this positive assessment and for the suggestions. We have now discussed our results in more detail regarding the context of T cell biology. Please find below how we included the suggested improvements in the revised manuscript.

Specific

comments:

1. Based on the data presented in Figure 1 D and E, it is clear there is depletion of IFNG and TNF mRNA 4hrs after activation and then the mRNA levels go up at 6h in both cases. However, the authors suggest that only TNF mRNA is depleted at 4hrs of activation (lines 169-172). The median number of IFNG mRNA gradually decreases after 1h of activation and reaches a low at 4h and then substantially increases by 6h. Did the authors measure gene expression of these mRNAs at later time points in the activation process? Perhaps transcription is coupled to mature mRNA levels in the cytoplasm and transcription is ramped up again once the cytoplasmic mRNA levels reach a lower threshold. Is this just an anomaly of the system or is gene expression pattern of cytokines upon T-cell activation cyclical?

We apologize for the lack of clarity. We will rephrase the text (**lines 149-154**) to discuss also the decrease in IFNG mRNA expression at 4hrs. Specifically, the mRNA expression represents well the protein production kinetics that were previously reported by single-cell ELISA (DOI: <https://doi.org/10.1073/pnas.1117194109>): TNF secretion typically occurs early (protein production can be detected between 30-60 min of activation), while IFNG production starts to be generated later, i.e. between 60-120 min of activation, dominating the cytokine production at later time points (i.e. 3-6h), a time when TNF production is receding. Interestingly, the above mentioned study suggests that IFNG displays a wave-like dynamic, with a secondary increase beginning around 6 hours post-stimulation. We indeed detect similar dual production kinetics for *IFNG* mRNA.

2. In data presented Figure 2 and Suppl Figure 2, the authors show correlation between dual cytokine expression and biallelic expression. However, not all dual cytokine expressing cells show bi-allelic expression of both cytokines. It will be useful to know what fraction of cells are biallelic for both genes. Since the experiment was done using two color smFISH, a scatter plot will cluster those dual expressor cells for both cytokines that are also bi-allelic for both genes. Extending this further would be to systematically address protein expression in the various combination of expression patterns. Combining smFISH with immunofluorescence will help address this. Overall, these results will be helpful in getting a better understanding of gene expression patterns during T-cell activation.

We agree, and we had therefore originally attempted to perform this analysis. However, because the number of cells displaying monoallelic expression that co-express both cytokine mRNAs is very low, statistically meaningful analysis was not possible. We therefore have refrained from including them.

We also agree with the reviewer that simultaneous mRNA and protein detection would be interesting. It was therefore what we originally aimed for. Pilot experiments, however, revealed that when the two techniques of detecting mRNA/protein are combined, primary T cells exhibit high background fluorescence that interferes with reliable mRNA and protein signal. Furthermore, quantification of protein production by Immunohistochemistry is impossible due to high levels (the signal is just bright). Another hurdle is the rapid secretion of IFN and TNF. Therefore, protein measurements are typically performed upon treatment with Brefeldin A. This, in turn, induces ER stress and thus interferes with mRNA production kinetics. We

therefore opted to measure mRNA and protein expression in parallel, which allowed for simultaneous semi-quantitative measurements of cytokine protein levels.

3. The mRNA localization data presented in Figure 3A and the associated supplemental figure: A better analysis and representation of the data presented in 3A would be a scatter plot of individual cells for their nuclear and cytoplasmic localization of mature mRNA. The authors might also want to extend this analysis based on the data presented in Figure 2 for dual expressors and bi-allelic expression. In other words, do cells with bi-allelic expression have more mRNA localized in the cytoplasm, and does this hold true in dual expressor cells? In the context of translation dependent decay of mRNA, do the dual expressor cells with biallelic expression fare better thereby producing and secreting cytokines continuously?

We now included the scatter plot as **NEW Supplementary Figure 3A**:

NEW Suppl. Fig. 3A Scatter plot of cytoplasmic mRNA (x-axis) versus nuclear mRNA (y-axis) per cell of Teff cells expressing ≥ 1 mature mRNA. $n \geq 500$ cells/time point, pooled from 3 donors. Black line: correlation line.

As to the question on the distribution of *IFNG* and *TNF* mRNA in bi- and mono-allelic cells, we had raised this question already ourselves. However, as the reviewer can appreciate from Supplementary Figure 1, the number of monoallelic cells is low, in particular for *IFNG*. This impedes any meaningful statistical analysis and interpretation. We therefore decided to omit this analysis.

4. The data presented for IFNG in Figure 4 is quite intriguing. In HuR-KO cells at 2h post induction, two of the three donors cell lines have only a small fraction of cells producing protein compared to the controls, however, they are substantially higher than the KO cells at time "0". Surprisingly, the amount of protein produced by these cells (panel B), although statistically lower than the control, is substantially higher than KO cells at

"0" h. Does the lone donor cell line with higher number of protein producing cells contribute to majority of the protein produced? There appears to be substantial difference between the three donor cell lines in the number of protein producing cells and mature IFNG mRNA after activation (Suppl Figure 1G & H). The authors may wish to compare the results before combining the data of all three donor cell lines before interpreting the data.

As suggested, we have re-examined the protein production data for individual donors (**NEW Supplementary Figure 4D**). Because the protein expression across donors was similar, we conclude that- even though there is some level of inter-donor variability- the protein production is not dominated by one donor. The observed inter-donor heterogeneity in mRNA and protein levels is a feature of primary human T cells and was previously reported ([10.1073/pnas.1117194109](https://doi.org/10.1073/pnas.1117194109); [10.4049/jimmunol.1601531](https://doi.org/10.4049/jimmunol.1601531)). We have clarified this point in the revised discussion (**lines 248-250**).

NEW Supplementary Figure 4D. Violin plot of protein fluorescence intensity per cell per each donor (D1, D2, D3) at different time points. Red bar: median expression.

5. Also intriguing, HuR knock out results in a significant increase in transcription of IFNG at time "0" (Figure 4, panel E). Despite this, there is a significant loss in transcription of IFNG 2h post activation. However, there is a significant accumulation of mature mRNA (panel D). Combined with the protein expression data presented in

panels A & B, and the fact that translation induces mRNA decay, how do the authors reconcile this data?

We thank the reviewer for this insightful comment and agree that a comprehensive analysis of transcription, mRNA, and protein levels is critical for understanding the impact of HuR in Teff cells. We have expanded the discussion (**lines 364-375**) accordingly and interpret the data as follows:

Our data indicate that HuR deletion enhances IFNG mRNA transcription under resting condition, suggesting a repressive role for HuR in regulating *IFNG* mRNA transcription. Upon stimulation, HuR-del Teff cells sustain elevated levels of both nascent and mature IFNG mRNA during the first hour compared to control Teff cells (**Fig. 4D, 4E**), yet fail to efficiently translate this mRNA into protein. The subsequent decline in IFNG transcription at 2h may reflect a feedback mechanism, where accumulated mRNA in the nucleus at 1h (**Fig. 5B**) triggers transcription repression, as reported in previous studies (Berry et al., 2022, DOI: [10.1016/j.cels.2022.04.005](https://doi.org/10.1016/j.cels.2022.04.005)).

6. The differential effect of HuR knock out on poly(A) tail length of IFNG and TNF mRNA is of great importance and the most striking finding in this study! It is generally accepted that poly(A) tail length contributes to mRNA stability and survival. The results presented in Figures 4 and 5 argue otherwise. Only a small fraction of TNF mRNA have full length poly(A) tails, however, the number of mature TNF mRNA in KO cells is much greater than the control even at "0" h. In addition, the TNF mRNA appear to be well translocated into the cytoplasm and effectively translated. Given these conflicting observations, what possible mechanism do the authors envision that can explain this result.

We agree that a more comprehensive discussion of the possible mechanisms regulating TNF mRNA and protein would be beneficial. We will include the following paragraph in the discussion (**lines 348-362**):

Intriguingly, even though HuR deletion perturbs the poly-adenylation of *TNF* mRNA, at the same time it results in increased mRNA abundance, efficient translocation and translation. This finding indicates that the poly(A) tail length for *TNF* is decoupled from mRNA stability and translational output. Of note, such decoupling is not unique to HuR depletion. Previous studies reported translationally active mRNAs, including *TNF*, in activated macrophages, even

though the poly-A tails were shortened (Kwak et al., 2022, doi: [10.1261/rna.078918.121](https://doi.org/10.1261/rna.078918.121), Crawford et al. 1997, DOI: [10.1074/jbc.272.34.21120](https://doi.org/10.1074/jbc.272.34.21120)). Whether this is a feature unique to immune cells, or whether it generally occurs in cells during rapid adaptations to external cues remains to be determined. The molecular cues that decouple polyA tail length from mRNA abundance and translation are also unknown. It is conceivable that secondary structures or RBPs interacting with *TNF* mRNA at specific locations impede the access to the degradation machinery and/or can retain *TNF* mRNA at the translation machinery. Irrespective of the mechanism, our findings support previous reports suggesting a functional decoupling (Kwak et al., 2022, doi: [10.1261/rna.078918.121](https://doi.org/10.1261/rna.078918.121), Crawford et al. 1997, DOI: [10.1074/jbc.272.34.21120](https://doi.org/10.1074/jbc.272.34.21120)), and demonstrate that the poly-A length is not a key predictor for mRNA abundance and translation of TNF in the context of immune activation.

Again, plotting the data presented in Figure 5A as a scatter plot between # of RNA in the cytosol vs nucleus will give a better picture of the localization changes in individual cells.

We thank the reviewer for this suggestion. We now provide a scatterplot of cytoplasm and nuclear mRNA in HuR-del and Control Teff cells in **NEW Supplementary Figure 5A**.

New Suppl. Fig. 5A Scatter plot of cytoplasmic mRNA (x-axis) versus nuclear mRNA (y-axis) per cell of Teff cells expressing ≥ 1 mature mRNA. $n \geq 500$ cells/time point, pooled from 3 donors. Yellow dots are Control Teff cells and green dots are HuR-del Teff cells. Yellow and green line: correlation line with correlation coefficient.

7. A more elaborate discussion of the results as it relates to the biology of cytokine gene expression during T-cell activation will immensely strengthen the manuscript.

We appreciate the reviewer's suggestions. As highlighted in our response to reviewer #1, we expanded the discussion, including the points discussed above.

Minor comment:

Full Revision

Images of cells with smFISH data (Figures 1, 3 & 4) must be bigger for better visualization. Show images with only a couple of cells enlarged to show the mRNA spots more visibly. Include images with more cells in the supplement instead.

We have increased the size of the smFISH images as much as we could while complying with the journal's figure size guidelines.

****Referee's cross-commenting****

I must confess I am not an immunologist, so my knowledge of the intricacies of gene expression in T-cells is very limited. However, I do have a fair sense of transcription regulation and use single molecule approaches, especially smFISH, to address these questions. I agree with the other reviewer the study is of significance, especially the advancement in the ability to do smFISH in primary cells, a challenge that I know first hand.

I also have to agree with the other reviewer that the discussion was too short and the authors shied away from the bigger picture of being able to comment on regulation of expression of cytokines during T-cell activation. It is remarkable that they see heterogeneity in gene expression of the individual target genes and bi-allelic expression. The other point of interest is the difference in p(A) tail length and its potential role in regulating TNF gene expression.

Reviewer #2 (Significance (Required)):

The successful implementation of a modified smFISH protocol used in this study is a welcome resource for all labs that want to study small human primary cells that are difficult to culture on glass coverslips and grow as suspension cultures.

Overall, this work is of high quality and can be better presented to fully explore and discuss the biological implications of the observations from the study. It is not clear to me if the authors wished to present this manuscript reporting an advancement in technology tool to study gene expression during T-cell activation, or a more in-depth study of gene expression.

Full Revision

The study will benefit the larger community that use single molecule approaches to understand gene expression.

We thank the reviewers for their positive assessment and are pleased that they recognize our study as a valuable methodological contribution to the field, in addition to providing novel biological insights. As noted above, we will expand the discussion to further emphasize the broader implications of our findings.

Dr. Monika C Wolkers
Sanquin Research
Hematopoiesis
Plesmanlaan 125
Amsterdam 1066CX
Netherlands

8th Aug 2025

Re: EMBOJ-2025-121882-T
Single molecule imaging of transcription dynamics, RNA localization and fate in T cells

Dear Dr. Wolkers,

Thank you again for submitting your revised Review Commons manuscript for consideration by The EMBO Journal. As discussed earlier, we decided to consider the work as a potential candidate for our Methods/Resource section, and therefore returned it directly to the original referees 1 and 2. I am happy to say that both consider the manuscript significantly improved and the original concerns well-answered. Referee 2 still argues for inclusion of a somewhat more in-depth discussion of the T-cell smFISH results in the context of cytokine biology, and from our side, we would require amendment of the work with a step-by-step protocol version to aid replicability and broad application of the method. Following these final modifications as well as adjustment to our specific journal format and editorial requirements (as follows), we should be able to proceed with acceptance and publication of the study:

GENERAL:

- Please download and complete our author checklist (link provided below).
- Please provide suggestions for a short 'blurb' text prefacing and summing up the conceptual aspect of the study in two sentences (max. 250 characters), followed by 3-5 one-sentence 'bullet points' with brief factual statements of key results of the paper; they will form the basis of an editor-written 'Synopsis' accompanying the online version of the article. Please also upload a synopsis image, which can be used as a "visual title" for the synopsis section of your paper (maybe based on a slimmed-down version of Figure 1?). The image should be in PNG or JPG format, and please make sure that it remains in the modest dimensions of (exactly) 550 pixels wide and 300-600 pixels high.
- You shall also receive a separate message from our Source Data curation team, with instructions on how to prepare and upload relevant image and numerical raw data.

TEXT:

- Please upload the manuscript text as an editable DOCX file, without marking of earlier changes, and without inclusion of figures.
- Please adjust the order as well as the headers of the different manuscript sections: Title page with complete author information, Abstract, Keywords, Introduction, Results, Discussion, Methods, Data Availability, Acknowledgements, Disclosure and Competing Interests Statement, References, Main Figure Legends, Tables, Expanded Figure Legends.
- On the abstract page of the manuscript, please include 4-5 general keyword terms to enhance searchability.
- Please note that Materials and Methods need to be described in the main text using our 'Structured Methods' format (for detail, see <https://www.embopress.org/page/journal/14693178/authorguide#structuredmethods>). The in-text "Methods" section should contain method and protocol descriptions (ideally using a step-by-step protocol format to facilitate adoption of the methodologies across labs), while all key reagents, experimental models, software and relevant equipment - including their sources and relevant identifiers - should be listed in a separately uploaded Reagents and Tools Table, a template for which can be downloaded from the above section of our Author Guidelines. Since we intend to publish this work in our Methods section, please make sure to also include a separate step-by-step protocol passage, which we further recommend posting protocols to a platform dedicated to structured protocols like protocols.io or any similar platform. For further information, please refer to <https://www.embopress.org/page/journal/14602075/authorguide#methodguide>

- As we are switching from a free-text author contribution statement towards a more formal statement based on Contributor Role Taxonomy (CRediT) terms, please remove the present Author Contribution section and instead specify each author's

contribution(s) directly in the Author Information page of our submission system during upload of the final manuscript. See <https://casrai.org/credit/> for more information.

- Please rename the Competing Interest section into "Disclosure and Competing Interests Statement", in accordance with our updated Guide to Authors (<https://www.embopress.org/competing-interests>)
- Please adjust the format of the reference list and of the in-text citations according to EMBO Journal format (alphabetical order, author name et al + year, first up to 10 authors should be listed, followed by 'et al' ...). Also carefully check all references for completeness, with citation year, volume, or page/locator numbers currently missing for several of them.
- Please rename the Data and Materials Availability section into "Data Availability".
- Please move the funding information, currently described in a separate section, into the Acknowledgements.

DATA:

- Please upload all main Figures as individual, image-only files with sufficient resolution/quality for production.
- The "Supplementary Material" file should be renamed to "Appendix", uploaded as PDF, and start with a title page containing a header ("Appendix for -xxx-") and a brief structured table of contents including page numbers. Contents should be renamed accordingly (both within the Appendix and when referencing them in the main text) into "Appendix Materials and Methods", "Appendix Figure S1/2/3...", and "Appendix Table S1/2/3...". Movies should not be mentioned in the Appendix. Finally, please rename the unusual M&M figures into proper Appendix Figures and reference them as such from the Appendix Methods section. To best organize this, I would suggest moving "Appendix Materials and Methods" to the end of the Appendix, and to create a new "Appendix Figure S6" containing the three Methods figures as panels A-C, respectively.
- Please convert all movies (whose files are currently missing!) into "Expanded View movies", adjusting the respective in-text callouts to "Movie EV1/2/3...", and moving each movie together with a text file containing its respective legend into a separate ZIP file before re-uploading.
- Finally, during routine pre-acceptance checks, our data editors have raised the following queries regarding figures, data, and legends, which I would ask you to address (ideally using the Track Changes option): Please provide the exact p values in the legends of figures 2A-D; 3F, 4A, B, D, E; 5A, B; S2 A, S4B, E

Should you need additional guidance/feedback regarding this final adjustments, please do not hesitate to contact us directly. Thank you again for the opportunity to consider this work for The EMBO Journal, and I look forward to receiving your final version!

With kind regards,

Hartmut

- 3) Revised manuscript text (including main tables, and figure legends for main and EV figures) has to be submitted as editable text file (e.g., .docx format). We encourage highlighting of changes (e.g., via text color) for the referees' reference.
- 4) Each main and each Expanded View (EV) figure should be uploaded as individual production-quality files (preferably in .eps, .tif, .jpg formats). For suggestions on figure preparation/layout, please refer to our Figure Preparation Guidelines: <http://bit.ly/EMBOPressFigurePreparationGuideline>
- 5) Point-by-point response letters should include the original referee comments in full together with your detailed responses to them (and to specific editor requests if applicable), and also be uploaded as editable (e.g., .docx) text files.
- 6) Please complete our Author Checklist, and make sure that information entered into the checklist is also reflected in the manuscript; the checklist will be available to readers as part of the Review Process File. A download link is found at the top of our Guide to Authors: embopress.org/page/journal/14602075/authorguide
- 7) All authors listed as (co-)corresponding need to deposit, in their respective author profiles in our submission system, a unique ORCID identifier linked to their name. Please see our Guide to Authors for detailed instructions.
- 8) Please note that supplementary information at EMBO Press has been superseded by the 'Expanded View' for inclusion of additional figures, tables, movies or datasets; with up to five EV Figures being typeset and directly accessible in the HTML version of the article. For details and guidance, please refer to: embopress.org/page/journal/14602075/authorguide#expandedview
- 9) To facilitate reproducibility and cross-laboratory adoption of methodologies, please structure the Materials & Methods section as outlined in our guide to authors, including a completed Reagents and Tools Table that can be downloaded from our author guidelines as well (<https://www.embopress.org/page/journal/14602075/authorguide#structuredmethods>).
- 10) Digital image enhancement is acceptable practice, as long as it accurately represents the original data and conforms to community standards. If a figure has been subjected to significant electronic manipulation, this must be clearly noted in the figure legend and/or the 'Materials and Methods' section. The editors reserve the right to request original versions of figures and the original images that were used to assemble the figure. Finally, we generally encourage uploading of numerical as well as gel/blot image source data; for details see: embopress.org/page/journal/14602075/authorguide#sourcedata

In the interest of ensuring the conceptual advance provided by the work, we recommend submitting a revision within 3 months (6th Nov 2025). Please discuss the revision progress ahead of this time with the editor if you require more time to complete the revisions. Use the link below to submit your revision:

Link Not Available

Referee #1:

The authors have clearly articulated the methodologies used to arrive at their results, and this helps enormously with clarity. They have also made sufficient effort to ensure that their work is put into a broader context for the field. This is an excellent piece of work that has broader implications, not just in terms of technical advance, but understanding details of biological mechanisms controlling cytokine production by immune cells.

Referee #2:

After reading the revised manuscript, it appears the authors have done a fairly good job of addressing the queries and comments. Yes, the manuscript is more aligned as a methods/resource paper, and the authors have done a significant job of dedicated optimization of existing protocols and successfully adapted it for cells growing in suspension. This is an advancement of significance and should work for suspension cells at large or be easily adaptable for a particular cell type. Moreover, the

authors clearly demonstrate they have established a method that retains cell morphology much better than other existing protocols, crucial for successful measurement of cytosolic mRNA by smFISH.

I wish the authors would have discussed their results in the context of cytokine biology.

Rev_Com_number: RC-2025-03025

New_manu_number: EMBOJ-2025-121882-T

Corr_author: Wolkers

Title: Single molecule imaging of transcription dynamics, RNA localization and fate in T cells

Rebuttal letter

Manuscript Number: EMBOJ-2025-121882-T

Corresponding Author(s): Monika C. Wolkers, Evelina Tutucci

Point by Point description of the revisions

Referee #1:

The authors have clearly articulated the methodologies used to arrive at their results, and this helps enormously with clarity. They have also made sufficient effort to ensure that their work is put into a broader context for the field. This is an excellent piece of work that has broader implications, not just in terms of technical advance, but understanding details of biological mechanisms controlling cytokine production by immune cells.

Referee #2:

After reading the revised manuscript, it appears the authors have done a fairly good job of addressing the queries and comments. Yes, the manuscript is more aligned as a methods/resource paper, and the authors have done a significant job of dedicated optimization of existing protocols and successfully adapted it for cells growing in suspension. This is an advancement of significance and should work for suspension cells at large or be easily adaptable for a particular cell type. Moreover, the authors clearly demonstrate they have established a method that retains cell morphology much better than other existing protocols, crucial for successful measurement of cytosolic mRNA by smFISH.

I wish the authors would have discussed their results in the context of cytokine biology.

We thank the reviewers for their evaluation and constructive feedback on our manuscript. In accordance with their recommendations, we have revised the discussion to provide a more comprehensive interpretation of our findings in the context of T cell biology (**lines 335–345**).

Dr. Monika C Wolkers
Sanquin Research
Hematopoiesis
Plesmanlaan 125
Amsterdam 1066CX
Netherlands

26th Sep 2025

Re: EMBOJ-2025-121882R
Single-molecule imaging of transcription dynamics, RNA localization and fate in human T cells

Dear Dr. Wolkers,

Thank you for submitting your final revised manuscript for our consideration. I am pleased to inform you that we have now accepted it for publication in The EMBO Journal.

Yours sincerely,

Hartmut Vodermaier
